# Identification of CD318, TSPAN8 and CD66c as target candidates for CAR T cell based immunotherapy of pancreatic adenocarcinoma

Daniel Schäfer[1,2,3], Stefan Tomiuk[1], Laura N. Küster[1], Wa'el Al Rawashdeh[1], Janina Henze[1,2,3],
German Tischler-Höhle[1], David J. Agorku[1], Janina Brauner[1], Cathrin Linnartz[1], Dominik Lock[1], Andrew Kaiser[1],
Christoph Herbel [1], Dominik Eckardt[1], Melina Lamorte[4], Dorothee Lenhard[4], Julia Schüler [4], Philipp Ströbel[5],
Jeannine Missbach-Guentner[3], Diana Pinkert-Leetsch[3,6], Frauke Alves [2,3,6], Andreas Bosio[1] & Olaf Hardt [1✉]

A major roadblock prohibiting effective cellular immunotherapy of pancreatic ductal adeno-carcinoma (PDAC) is the lack of suitable tumor-specific antigens. To address this challenge, here we combine flow cytometry screenings, bioinformatic expression analyses and a cyclic immunofluorescence platform. We identify CLA, CD66c, CD318 and TSPAN8 as target candidates among 371 antigens and generate 32 CARs specific for these molecules. CAR T cell activity is evaluated in vitro based on target cell lysis, T cell activation and cytokine release. Promising constructs are evaluated in vivo. CAR T cells specific for CD66c, CD318 and TSPAN8 demonstrate efficacies ranging from stabilized disease to complete tumor eradication with CD318 followed by TSPAN8 being the most promising candidates for clinical translation based on functionality and predicted safety profiles. This study reveals potential target candidates for CAR T cell based immunotherapy of PDAC together with a functional set of CAR constructs specific for these molecules.

[1] Miltenyi Biotec GmbH, R&D, Bergisch Gladbach, North Rhine-Westphalia, Germany. [2] University Medical Center Göttingen, Clinic for Hematology and Medical Oncology, Göttingen, Lower Saxony, Germany. [3] University Medical Center Göttingen, Institute for Diagnostic and Interventional Radiology, Göttingen, Lower Saxony, Germany. [4] Charles River Discovery Research Services GmbH, Freiburg, Baden-Wuerttemberg, Germany. [5] University Medical Center Göttingen, Institute for Pathology, Göttingen, Lower Saxony, Germany. [6] Max Planck Institute for Experimental Medicine, Translational Molecular Imaging, Göttingen, Lower Saxony, Germany. ✉email: olaf@miltenyi.com

Pancreatic cancer is a devastating disease. The 5-year overall survival rates have merely changed for the past decades and it is currently the fourth leading cause of cancer-related deaths in Western countries[1,2]. Surgery still is the only potentially curative treatment, but only around 20% of patients show a resectable disease stage at diagnosis[3]. Median overall survival with state-of-the-art treatment ranges from 26 month for patients with resectable disease to less than 6 months when already metastatic[4]. Thus, there is an unmet need for new therapeutic options. A new and promising therapeutic approach is chimeric antigen receptor (CAR) T cells. CAR T cells showed unprecedented efficacy in the treatment of B-cell malignancies[5,6]. They typically rely on pan-B cell antigens such as CD19 or CD20 and do not discriminate between healthy and tumor cells. As a consequence, all B cells are depleted, which is regarded as an acceptable side effect since it is otherwise well tolerated[7]. This is not the case for almost all target antigens in solid tumors until today and remains one of the central problems of solid tumor CAR T cell therapies. Prominent candidates among the targets which are currently under investigation in clinical trials for pancreatic cancer are carcinoembryonic antigen (CEA), human epidermal growth factor receptor 2 (HER2), mucin 1 (MUC1), prostate stem cell antigen (PSCA), prominin 1 (PROM1), epidermal growth factor receptor (EGFR), and mesothelin (MSN)[9]. These target candidates all have in common their shared expression on malignant and healthy tissues[8,10–15] and toxicities in humans were already reported for HER2 and CEA[16,17]. While administration of HER2-specific CAR T cells ended fatal for the patient[16], the use of CAR T cells against CEA caused only mild toxicities and also very limited efficacy[17]. These examples underline how important the aspect of safety is, which in case of CAR T cells comes with tumor specificity and off-tumor expression in dispensable cell types only.

Until today, a broad and systematic target antigen screen for CAR T cell therapy of pancreatic cancer that compares the specificity of a multitude of target candidates and their off-tumor expression has not been reported. Likewise, empirical studies needed to determine an optimal CAR design on suitable targets for this disease are also scarce[18].

Hence, we aim to close this gap and present here a systematic approach for CAR target screening that first narrows down the field of target candidates from 371 to 50 by flow cytometric analysis of 17 pancreatic cancer patient-derived xenograft (PDX) models. We investigate further the RNA and protein expression profiles of these target candidates, which are available in public online data banks. We rank the candidates depending on their expression in different healthy tissues and cell types. In addition, we examine the expression of a multitude of these target candidates within the primary pancreatic cancer tissues from patients using a cyclic immunofluorescence (IF) imaging platform. This technique enables to survey expression profiles of several dozens of antigens on the very same tissue section. We finally verify these results using flow cytometry on seven additional primary PDACs as well as on three metastatic lesions. Based on these results, we design 32 CARs specific for the four most promising target candidates, CLA, CD66c, TSPAN8, and CD318 with varying spacer lengths and scFv orientations. We empirically evaluate the CAR constructs in terms of cytotoxicity, cytokine release, and cell phenotype profile. CAR constructs that perform best in vitro are then examined in two independent preclinical mouse xenograft models and evaluated for their expression on healthy tissues by cyclic immune fluorescence and flow cytometry resulting in promising candidates for future clinical trials.

## Results

### Identification of PDAC cell surface target candidates for CAR T cell-based immunotherapy

As pancreatic ductal adenocarcinoma (PDAC) attributes to around 85% of all pancreatic cancer cases, we decided to use PDX models of PDAC for initial candidate identification due to their good availability and proven predictivity for the disease[19]. A scheme of the workflow for identification of target candidates applied in this study is depicted in Supplementary Fig. 1. Overall, we analyzed 17 independent PDX models representing 15 different mutational backgrounds (Supplementary Data 1). Initially, we screened two PDX models representative of PDAC concerning histology, mutational profile, and characteristic response to standard-of-care drugs (Charles River, personal communication, 2016) using a commercially available antibody array containing antibodies specific for 371 surface antigens, including antigens already under clinical investigation for CAR T cell-based treatment of PDAC, such as HER2, MUC1, PROM1, and CEA. We found 105 antigens to be expressed on more than 10% of the PDX cells on at least one of the PDX. We then used antibodies specific for these 105 antigens and measured their expression on two additional representative PDX models followed by a manual exclusion of non-suitable target candidates, such as HLA molecules which were present in the pre-set screening plates. We measured the remaining 50 surface antigens, which were expressed in at least 20% of all tumor cells of at least three out of four PDX models, on 13 more xenografts (Fig. 1a and Supplementary Fig. 2). Remarkably, MUC1 and HER2 did not match these criteria. A family of proteins that was expressed on many tumor cells of the PDX models were the tetraspanins. Prominently expressed members were CD9, CD63, CD82, and CD151. Another family that showed expression on many tumor cells throughout the different PDX models was the CEA family. Its members CEACAM1, CEACAM3, CEACAM5, and CEACAM6 could all be recognized by the pan-reactive CD66acde antibody. However, the expression pattern of CD66acde was paralleled by the expression of CD66c alone, and based on the higher specificity of the single molecule binder we chose this one for further evaluation. Unexpectedly, we found the cutaneous lymphocyte antigen (CLA) expressed on all but one xenograft. CLA is described as a binding epitope of the antibody clone HECA-452 and includes a Sialyl-Lewis[x] glycan structure[20,21]. So far, CLA was only observed to be expressed on subsets of leukocytes[20,22–25]. This is, in two ways, an interesting finding: (1) it was never described before to be expressed on pancreatic cancer cells and (2) was so far only reported to be expressed on subsets of certain cell types, which could mean no essential tissues may be harmed when targeted by CAR T cells.

### CD66c, CD318, TSPAN8, and CLA exhibited restricted off-tumor expression in human tissues

Next, we prioritized the 50 surface structures from our antibody screen with respect to their off-tumor expression. We assigned the corresponding genes to their respective antigens. For some instances, multiple genes had to be assigned to a single antibody. For example, CD66acde represents CEACAM1, CEACAM3, CEACAM5, and CEACAM6. In cases like this, each gene was investigated independently. In case an antibody was specific for a glycostructure, we assigned the respective backbone protein to it (for example, CLA can be a glycostructure on SELPLG[20]), if possible. Subsequently, we extracted RNA and protein expression data of the assigned genes from the following data sources: Human Protein Atlas[26], ProteomicsDB[27], Human Proteome Map[28], and GTEx. Next, we defined rankings independently for each dataset. Detailed information about the ranking procedure can be found in the Methods section. In brief, genes of each dataset were ranked by the total

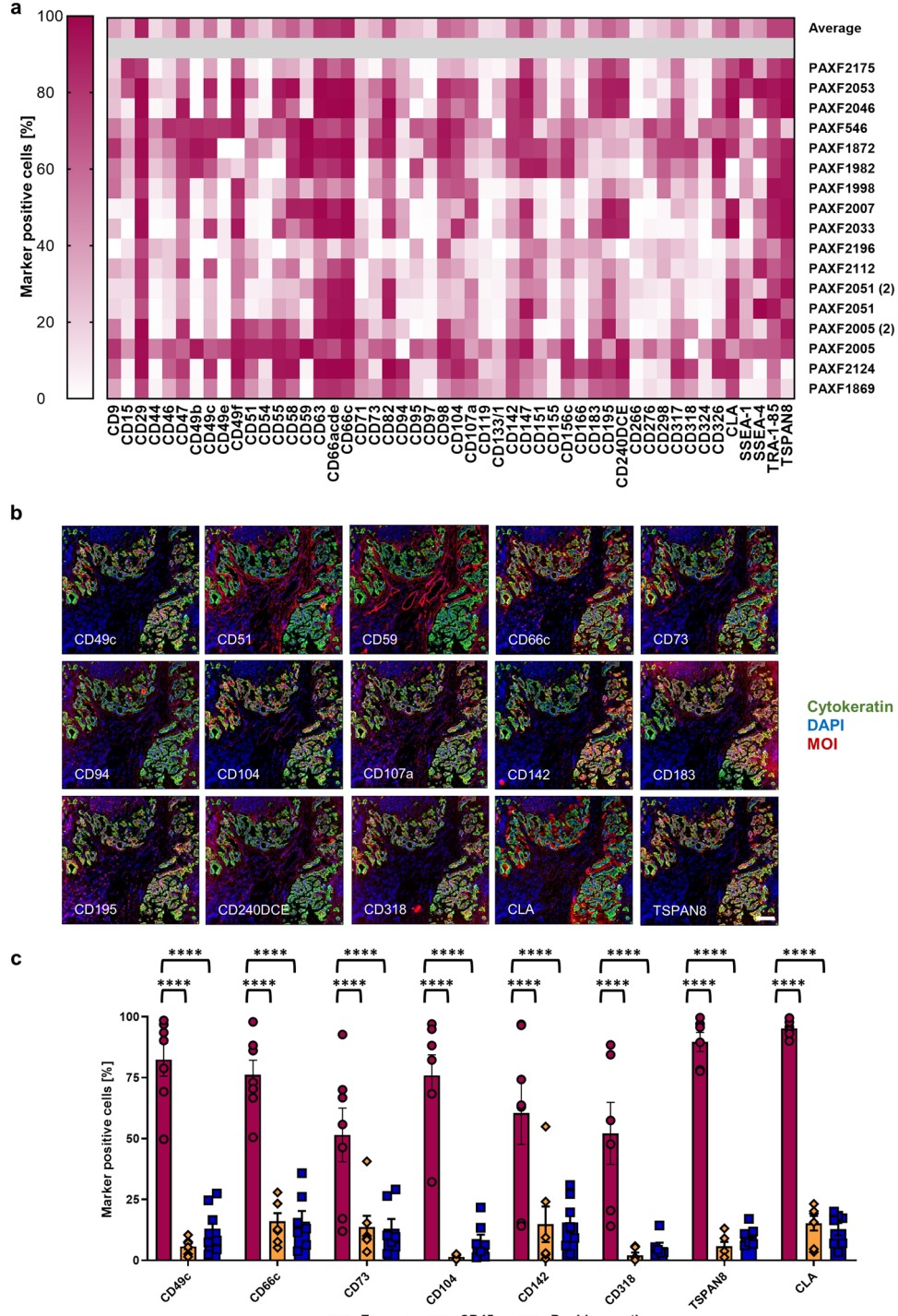

**Fig. 1 Identification of PDAC CAR target candidates. a** Top 50 antigens expressed on the cell surface of PDAC PDX models as determined by flow cytometry-based screening of a 371 monoclonal antibody panel. **b** Selection of 15 cyclic immunofluorescence images of a representative human PDAC tissue to evaluate co-expression of the marker of interest (MOI) with cytokeratin-positive tumor cells. Images are representative for two cyclic IF runs on two different PDAC specimen. **c** Flow cytometric analysis of expression for eight target candidates on primary human PDAC samples. Data represent mean ± s.e.m. ($n = 7$, ****$P < 0.0001$ determined through two-way ANOVA with Tukey's test). Scale bar = 100 μm. Source data are provided as a Source Data file.

number of tissues/cell types it was expressed in, as well as their overall expression levels. Subsequently, we calculated a rank sum that mirrors the overall expression throughout all data sources (Supplementary Data 2 and Supplementary Fig. 3). The final rank was then calculated based on the quotient of the rank sum of the target candidate divided by the number of data sources it was

found in. As a consequence, target genes expressed in a low number of tissues were prioritized as they are expected to cause lower potential off-tumor effects.

All four members of the CEA family that are bound by the CD66acde antibody exhibited only restricted expression over several tissues (Supplementary Data 2 and Supplementary Fig. 3).

Other target candidates that appeared top-ranked were the Rh blood group polypeptides RHD and RHCE (CD240DCE), the C–C chemokine receptor type 5 (CCR5, CD195), CXC chemokine receptor 3 (CXCR3, CD183), or CUB domain-containing protein 1 (CDCP1, CD318). Remarkably, from the five tetra-spanins found prominently expressed on the xenografts, only TSPAN8 remained in the 20 top-ranked candidates, showing that TSPAN8 has the most restricted expression profile of them. Furthermore, we found SELPLG as the protein backbone of CLA also belonging to the 20 best-ranked candidates.

In addition, we included MSN and EGFR into our bioinformatical analysis as they were not present in the antibody library, whereas other candidates currently evaluated in clinical trials, such as CEA, HER2, CD133, and MUC1 were included in our screen, showing rankings similar to our subsequent target candidates (Supplementary Data 2 and Supplementary Fig. 3). Lowering RNA sequencing-based detection thresholds from 1 FPKM/TPM to 0.5 FPKM/TPM did hardly change the final ranking (Supplementary Data 3 and 4). As an independent validation of our bioinformatics-based strategy, we applied an algorithm according to the one developed by Perna et al.[29] resulting in a comparable prioritization (Supplementary Data 2). In an additional ranking, we split among "essential" (Supplementary Data 3) and "non-essential" tissues (Supplementary Data 4). We followed the categorization into essential and non-essential organs as proposed by Perna et al.[29] Also in this case, the overall ranking of our target candidates did not change considerably (only ±1 rank). Interestingly, established targets such as MSN moved down to a greater extent with regard to non-essential tissues ranking (−6 for MSLN) indicating a good level of confidence for the proposed candidates.

**Validation of CD66c, CD318, TSPAN8, and CLA expression and specificity**. An important information that is lost during the processing of samples for RNA or protein expression analysis is the spatial distribution of target candidate expression within the tissue. Strong mRNA expression values derived from small populations of cells that may be acceptable with respect to toxicities are intermingled with the rest and turn suitable targets to false negatives. To overcome these problems and to gain a better understanding of the target candidate expression in primary PDACs in situ, we used a cyclic IF imaging platform. This system operates by iterative fluorescence staining, imaging, and signal erasure, enabling the operator to identify the expression of dozens of targets on the very same tissue section. We performed two runs of cyclic IF imaging on two different human PDAC tumor tissues with 107 and 98 markers, respectively (Supplementary Data 5). These markers were a selection of the ones used on the PDX models plus markers to dissect known cell lineages. Our observations were centered on identifying candidates showing colocalization with tumor cells and no or low expression on non-tumor cells. Target candidates that were ranked promising through bioinformatics and showed specific expression on tumor cells from patients were CLA, TSPAN8, CD318, and CD66c (Fig. 1b, Supplementary Data 2–4, and Supplementary Fig. 3). Other candidates such as CD240DCE, CD195, and CD183, which were ranked high before by bioinformatics, exhibited strong expression on non-tumor cells. Some candidates appeared to be specific to the tumor cells, although they did not belong to the group of high-ranked targets, such as CD49c, CD73, CD104, and CD142 suggesting a generic epithelial reactivity (Fig. 1b). CD51, CD59, and CD107a had low bioinformatical rank and presented massive expression offside of tumor cells, nicely confirming the value of this approach.

These findings led to the decision to further verify the specificity of CD49c, CD66c, CD73, CD104, CD142, CD318, CLA, and TSPAN8 on primary pancreatic tumor cells using flow cytometry. We dissociated seven human PDAC specimens and gated either on tumor cells (EpCAM$^+$), leukocytes (CD45$^+$), or other cells (double negative). It was shown that these eight candidates indeed showed a significant enrichment of expression on tumor cells (Fig. 1c and Supplementary Fig. 4a). As bioinformatical rankings indicated a higher safety risk for CD49c, CD73, CD104, and CD142 (Supplementary Data 2–4 and Supplementary Fig. 3), we decided to continue with the validation of CLA, CD66c, CD318, and TSPAN8.

Next, we evaluated target candidate expression on human PDAC metastases, as CAR T cell therapy most likely will be initially applied in a metastatic stage. We could corroborate the expression of CD66c, CD318, and TSPAN8 on three primary tumors and their respective liver or lymph node metastases (Fig. 2a). In addition, we found all three target candidates and the protein backbone of CLA also expressed in a PDAC dataset including a metastasis generated by the TCGA Research Network (https://www.cancer.gov/tcga; Supplementary Data 6). With respect to subsequent preclinical evaluation, we assessed if target expression on cell line-derived xenografts (CDX) from AsPC1 and BxPC3 cells and on PDX (PAXF 1872) varied among orthotopically (o.t.) or subcutaneously (s.c.) implanted tumors (Fig. 2b and Supplementary Figs. 4b and 5). We found CD66c, CD318, and TSPAN8 equally expressed among our s.c. and o.t. transplanted CDX and PDX models using flow cytometry, IF, and immunohistochemistry (IHC) microscopy, indicating that s.c. and o.t. in vivo models represent valid options for later CAR T cell evaluation. Of note, metastatic lesions of CDX models resembled expression of primary tumors in most cases (Supplementary Fig. 4b). Unfortunately, we could not detect CLA on the matched tumor–metastases pairs or o.t. transplanted CDX, probably due to the lack of suitable antibodies for staining of FFPE tissue.

**CLA-specific CARs are dysfunctional, while CARs specific for CD66c, CD318, and TSPAN8 induce potent activation, cytokine release, and antitumor efficacy in vitro**. After identification of the four target candidates CLA, CD66c, CD318, and TSPAN8, we designed a CAR library with varying spacer lengths (XS, S, M, and L) and scFv orientations ($V_h$-$V_l$ and $V_l$-$V_h$) (Fig. 3a). It has been shown that these elements are critical for CAR function[18,30]. All CARs shared a second-generation design with CD3ζ and 4-1BB as stimulatory domains and a CD8α transmembrane domain.

First, the general ability of the CAR constructs to be expressed on the cell surface was assessed. Most M-sized CARs showed substantial mal-expression, which led us to exclude them from further analysis (Supplementary Fig. 6). Subsequently, we used an in vitro co-culture system to validate and rank functionality of the remaining CARs using cytotoxicity (Supplementary Movies 1 and 2), cytokine secretion, and activation marker upregulation as readout. We considered GM-CSF, IL-2, TNF-α, and INF-γ as the cytokines which have the highest value in terms of CAR T cell activity. The examined T cell markers were PD-1, LAG-3, 4-1BB, TIM-3, and LAG-3/4-1BB co-expression. To ensure inter-experiment comparability, we normalized the values to the highest of each assay. All CAR T cells were evaluated for unspecific activation and induction of tonic signaling in a co-culture with HEK293T cells, not expressing any of the target antigens (Supplementary Fig. 7a). Furthermore, general activation ability was tested by stimulation of T cells with PMA/Ionomycin (Supplementary Fig. 7b and Supplementary Data 7).

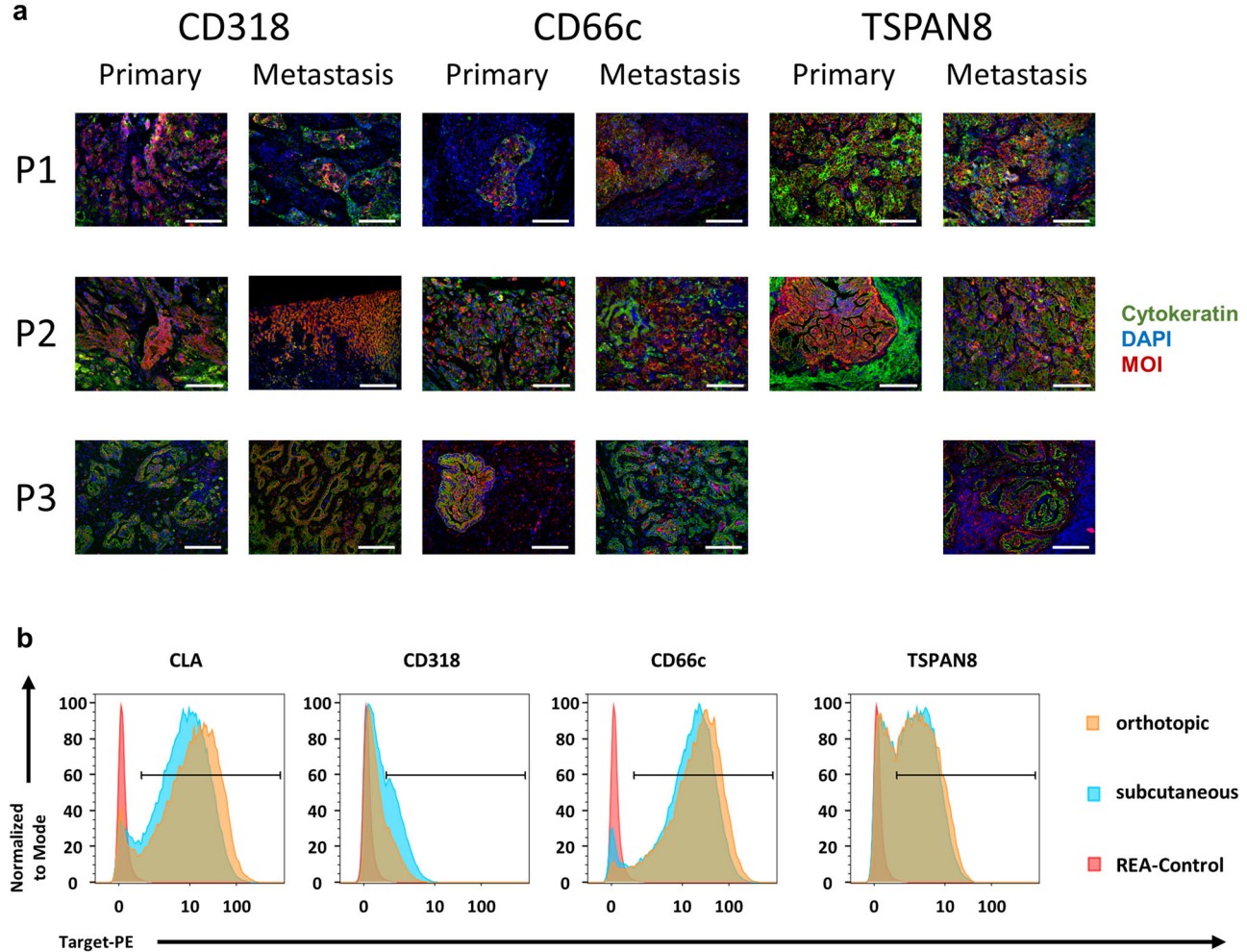

**Fig. 2 Target expression is preserved in human metastases and upon subcutaneous or orthotopic transplantation. a** Target candidate staining of three independent primary PDACs and matched metastases. Cell nuclei were stained with DAPI (blue), tumor cells by cytokeratin (green), marker of interest (MOI) is depicted in red. P1 and P3 metastases are of lymph node origin, P2 is liver derived. Images are representative for one stained sample. **b** Flow cytometry staining of target candidates on dissociated PAXF1872 PDX cells transplanted either orthotopically (blue) or subcutaneously (yellow). Respective isotype control is indicated in red. Scale bar = 150 μm for all tissues except for P2, metastasis, CD318 = 300 μm. P = patient. TSPAN8 staining of primary PDAC from P3 not available due to technical failure.

For CD66c CARs, the S and XS spacer performed best, the $V_l$-$V_h$ scFv orientation being superior over the $V_h$-$V_l$ orientation. The CD318 CARs featured similar behavior but the $V_h$-$V_l$ conformation appeared to be more functional. These findings were contrasted by the CARs specific for TSPAN8, for which the long spacer exhibited better cytotoxicity. All anti-CLA CARs performed much worse as compared to their counterparts for the other antigens (Supplementary Data 7 and Supplementary Fig. 8a). They displayed only weak killing and almost no cytokine release or marker upregulation. We think the cause of this was CLA to be a T cell self-antigen upregulated upon activation (Supplementary Fig. 9). As shown above (Supplementary Fig. 6), the expression levels of CLA-specific CARs in HEK293T cells were very low. This in combination with self-antigenicity may lead to binding of CAR and target already intracellular, and in consequence to degradation of this complex. The results of all in vitro functionality assays can be found in Supplementary Data 7. Finally, we directly compared the best-performing CARs based on the highest values of cytotoxicity, cytokine release, and marker upregulation (Fig. 3b and Supplementary Data 8). The CD66c S spacer CAR outperformed the XS spacer CAR. Both CD318 CARs showed fast and efficient target cell killing with

almost complete lysis using BxPC3 and the fastest kinetic using AsPC1 cells (Fig. 3c, d and Supplementary Fig. 8b, c). A slight difference in performance only appeared based on cytokine release and activation marker upregulation. The XS spacer released less cytokines but exhibited improved activation marker upregulation. All other CARs showed cell lysis between 40% and 60% within the first 48 h following an outgrowth control until 160 h (Fig. 3c and Supplementary Fig. 8b). The highest activation of TSPAN8-specific CARs showed the S spacer in $V_h$-$V_l$ orientation (Fig. 3b). As a consequence, we chose to evaluate the CAR constructs CD66c S $V_l$-$V_h$, CD318 XS $V_h$-$V_l$, and TSPAN8 S and L $V_h$-$V_l$ for in vivo functionality in two preclinical models.

**CAR T cells specific for the target candidates exhibit potent anti-tumor responses in vivo.** To evaluate the most promising CAR candidates in a preclinical setting, we engrafted Luc$^+$ AsPC1 cells s.c. in NSG mice. Tumor growth kinetics were evaluated by caliper and bioluminescence (BLI) measurements (Fig. 4a). After the first tumors reached a size of 25 mm², we injected $5 \times 10^6$ CAR T cells or untransduced T cells intravenously. Mock T cells exhibited no therapeutic benefit over the

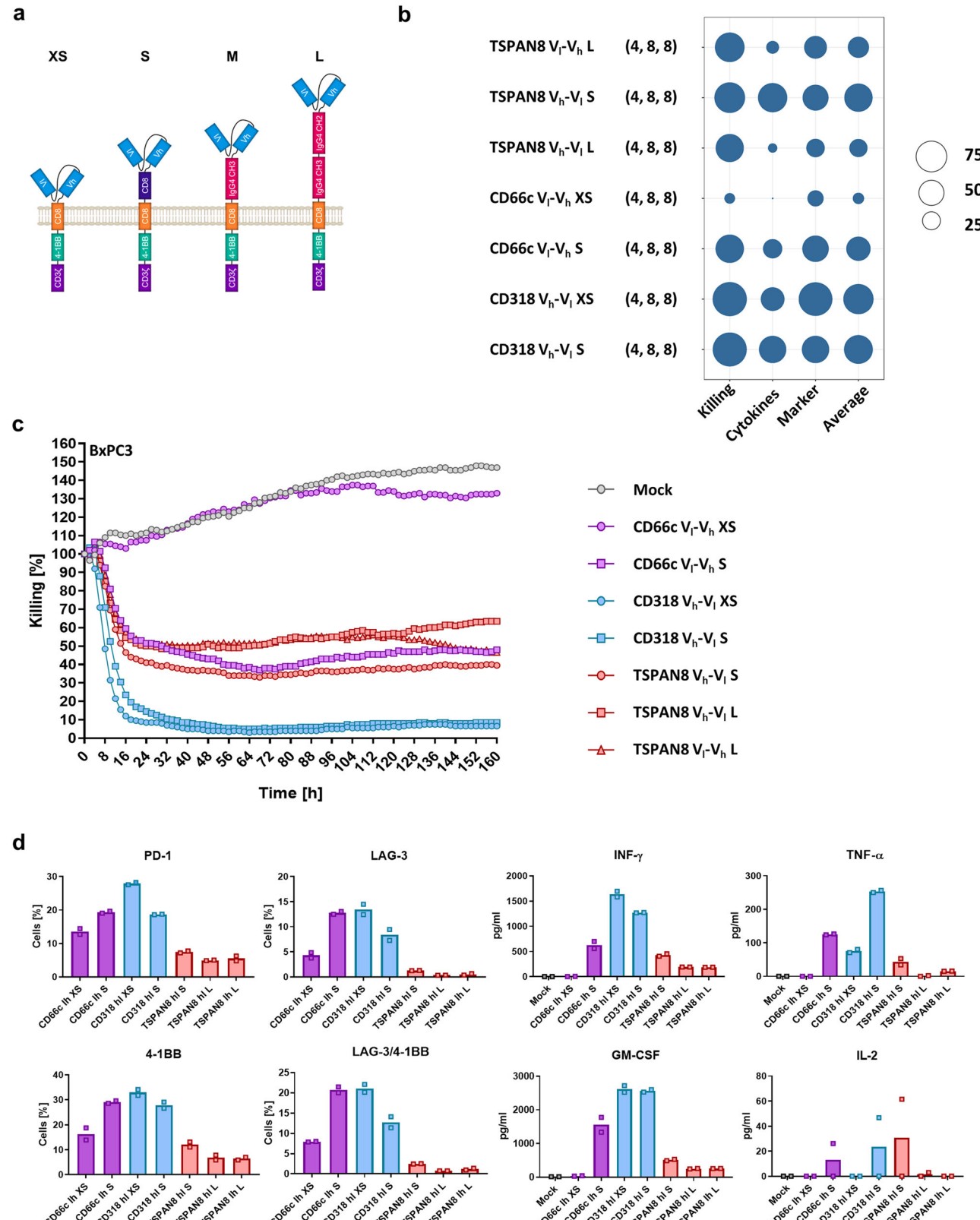

untreated group (Supplementary Fig. 10a). All the mice treated with Mock T cells were sacrificed latest 13 days post T cell injection due to appearance of ulceration in tumors (Supplementary Fig. 10b). Remarkably, all tumors irrespective of the treatment grew in terms of size until 9 days post T cell injection, although for some animals the BLI signal had decreased already,

after 6 days (Fig. 4b–d). We could verify that this drop in BLI signal and increase in size was accompanied by tumor cell lysis correlated with massive CAR T cell and macrophage infiltration (Fig. 5c). Complete and fast tumor eradication was observed in the CD318 XS CAR-treated group. The TSPAN8 S CAR showed a slower therapeutic kinetic, but achieved tumor clearance in three

**Fig. 3 Generation of target candidate-specific CAR constructs and evaluation of in vitro functionality. a** Scheme of the generation of our CAR library. The combination of four backbones differing in spacer length with two scFv orientations and four target candidate specificities resulted in 32 constructs. **b** Average relative target cell killing, cytokine release, marker upregulation, and overall performance (displayed as circle size) of the selected best-performing CARs from the initial screening assays (y axis). The number of replicates is indicated in parentheses behind the construct name (first value = n of killing assays, second value = n of cytokine release assays, third value = n of marker upregulation measurements). **c** Representative result for the kinetics of BxPC3 target cell killing by the selected CAR constructs. **d** Representative results showing cytokine release and activation marker expression patterns upon co-culture with BxPC3 target cells. Shown are mean ± s.e.m. (n = 2). Activation marker expression was measured at end point of cytotoxicity assay, cytokine release patterns were measured after 48 h. Source data are provided as a Source Data file.

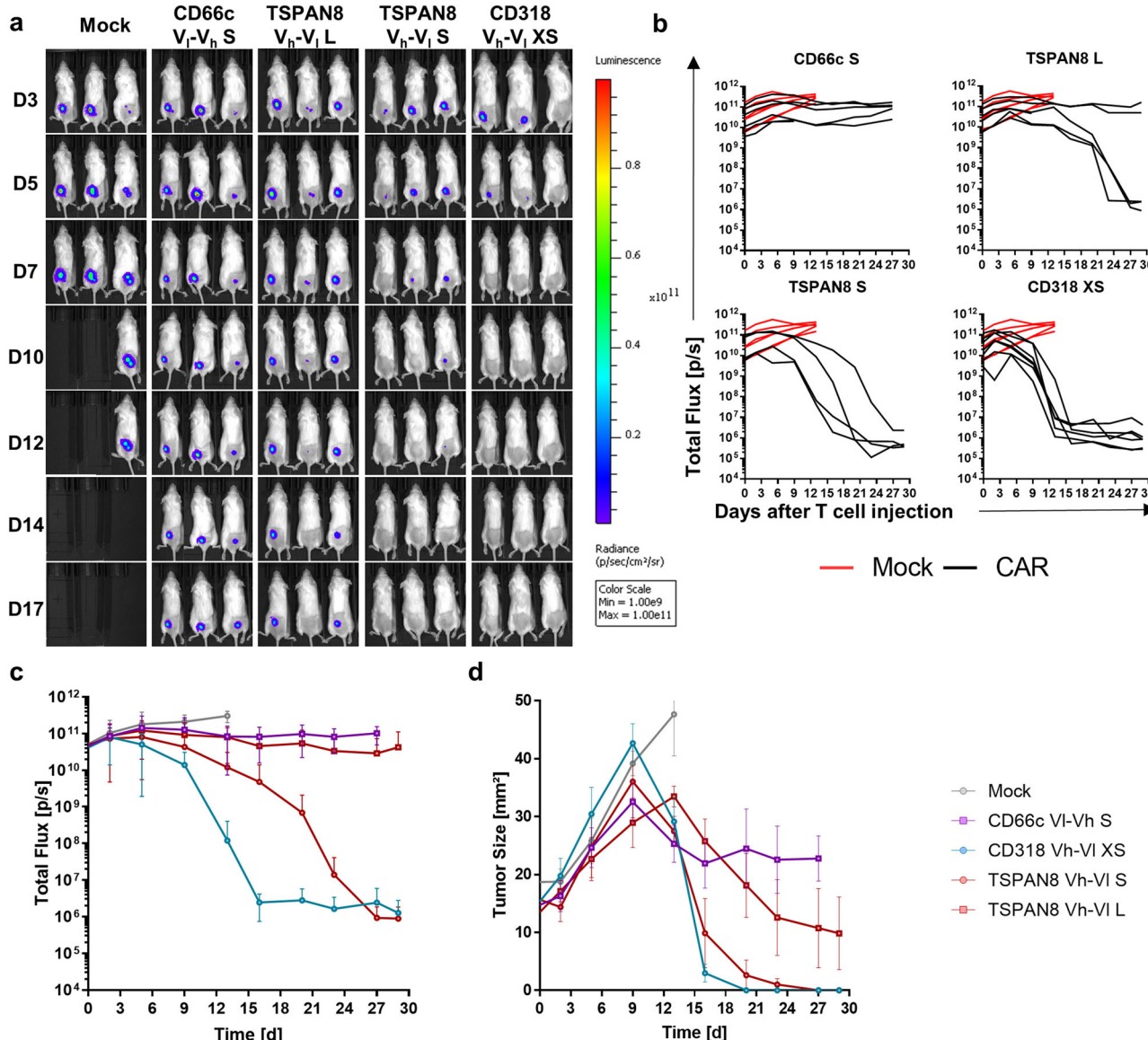

**Fig. 4 Evaluation of CAR T cell in vivo functionality in an AsPC1 xenograft model. a** Representative bioluminescence images of tumor-bearing NSG mice. Tumors were induced by subcutaneously transplanting luciferase expressing AsPC1 cells (color scale for all images, min = 1 × 10^9, max = 1 × 10^11). Mice were randomized and treated upon established solid tumors reached 25 mm² (day 7) by intravenous infusion of 5 × 10^6 CAR T cells or Mock T cells. **b** Development of tumor burden for individual mice treated either with Mock T cells or with the respective CAR T cells (Mock: n = 6, CD66c S: n = 5, CD318 XS: n = 7, TSPAN8 S: n = 4, TSPAN8 L: n = 6). **c** Average bioluminescence signal ± s.e.m. of the respective treatment groups (n equal to **b**). **d** Average tumor size ± s.e.m. of the respective treatment groups (n equal to **b**). Source data are provided as a Source Data file.

mice and almost complete clearance in one mouse. The TSPAN8 L CAR exhibited heterogeneous response with three complete responders and two mice showing stable tumor burden. The CD66c S spacer performed poorly in comparison but still stabilized the tumor burden in all animals (Fig. 4b, c). Flow cytometric analysis of spleen CAR T cells revealed that the groups CD318 XS

and TSPAN8 S had the highest CAR T cell counts at the end of the experiment correlating with the highest anti-tumor activity (Fig. 5a). In addition, the two low-responder mice showed the lowest CAR T cell count inside the TSPAN8 L group. However, the comparable cell counts of the CD66c group showed that the CAR T cell number alone is not sufficient to explain the efficacy.

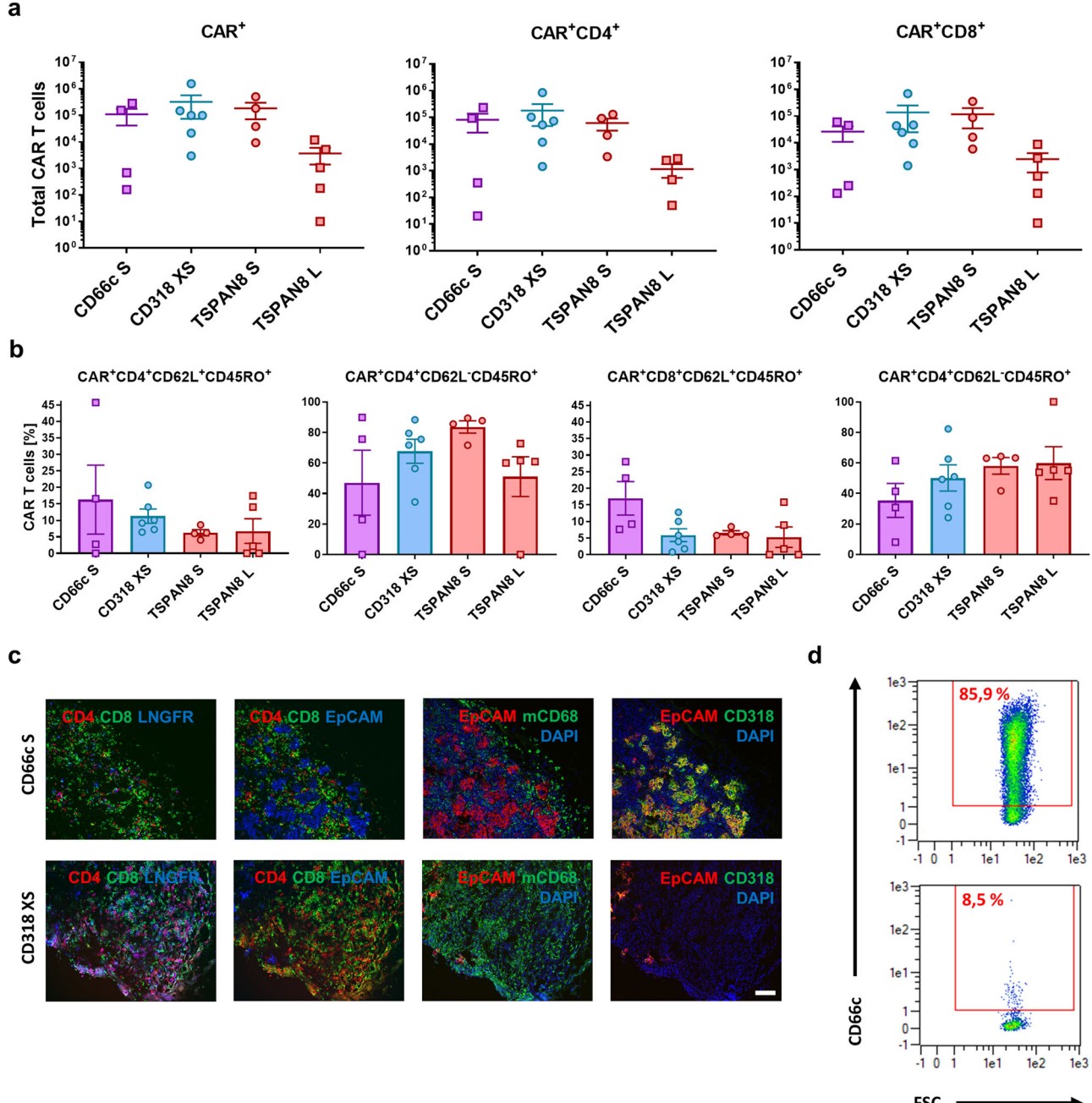

**Fig. 5 Ex vivo analysis of the T cell phenotype and AsPC1-derived tumor tissues upon treatment with CAR T cells. a** Number of CAR T cells in the spleen at the end of the experiment 27 days post CAR T cell injection (CD66c S: $n = 4$, CD318 XS: $n = 6$, TSPAN8 S: $n = 4$, TSPAN8 L: $n = 5$). **b** Phenotype of CAR T cells in the spleen at the end of the experiment 27 days post CAR T cell injection, as demonstrated by the percentage of $T_{CM}$ and $T_{EM}$ ($n$ equal to **a**). All data are shown as mean ± s.e.m. **c** Representative immunofluorescence images of (CAR) T cell tumor infiltration, macrophage tumor infiltration and target expression (CD318 XS tumor 9 days post CAR T cell injection, CD66c S tumor 27 days post CAR T cell injection). Staining was performed on one tumor of the respective treatment group and each image is representative for at least two regions of interest. Regions of interest during cyclic IF were chosen based on manual prestaining of DAPI and EpCAM. **d** Density plots of a dissociated AsPC1 xenograft showing CD66c expression 35 days post injection of $5e^6$ CD66c S $V_l$-$V_h$ CAR T cells (top) and the unstained control (bottom). Scale bar = 100 μm. Source data are provided as a Source Data file.

Flow cytometric and microscopy analyses of the tumors ex vivo showed that unresponsiveness was not linked to target down-regulation (Fig. 5c, d and Supplementary Fig. 5). Mice treated with CD318 CARs showed increased infiltration of T cells, CAR T cells, and macrophages as compared to those treated with CD66c CAR T cells, whereas the composition of central memory ($T_{CM}$) and effector memory ($T_{EM}$) phenotypes did not differ significantly among the groups (Fig. 5b).

To confirm the robustness of our results, we challenged these CAR constructs in a second preclinical setting using BxPC3 and an increased dose $1 \times 10^7$ CAR T cells (Fig. 6a–c). As seen previously, Mock T cells lacked therapeutic effect (Supplementary Fig. 10). The CD318 XS CAR again achieved a complete tumor eradication showing a fast kinetic. TSPAN8 S CARs showed a higher therapeutic functionality as the L spacer but both only induced stable tumor burden. In this model, no therapeutic effect

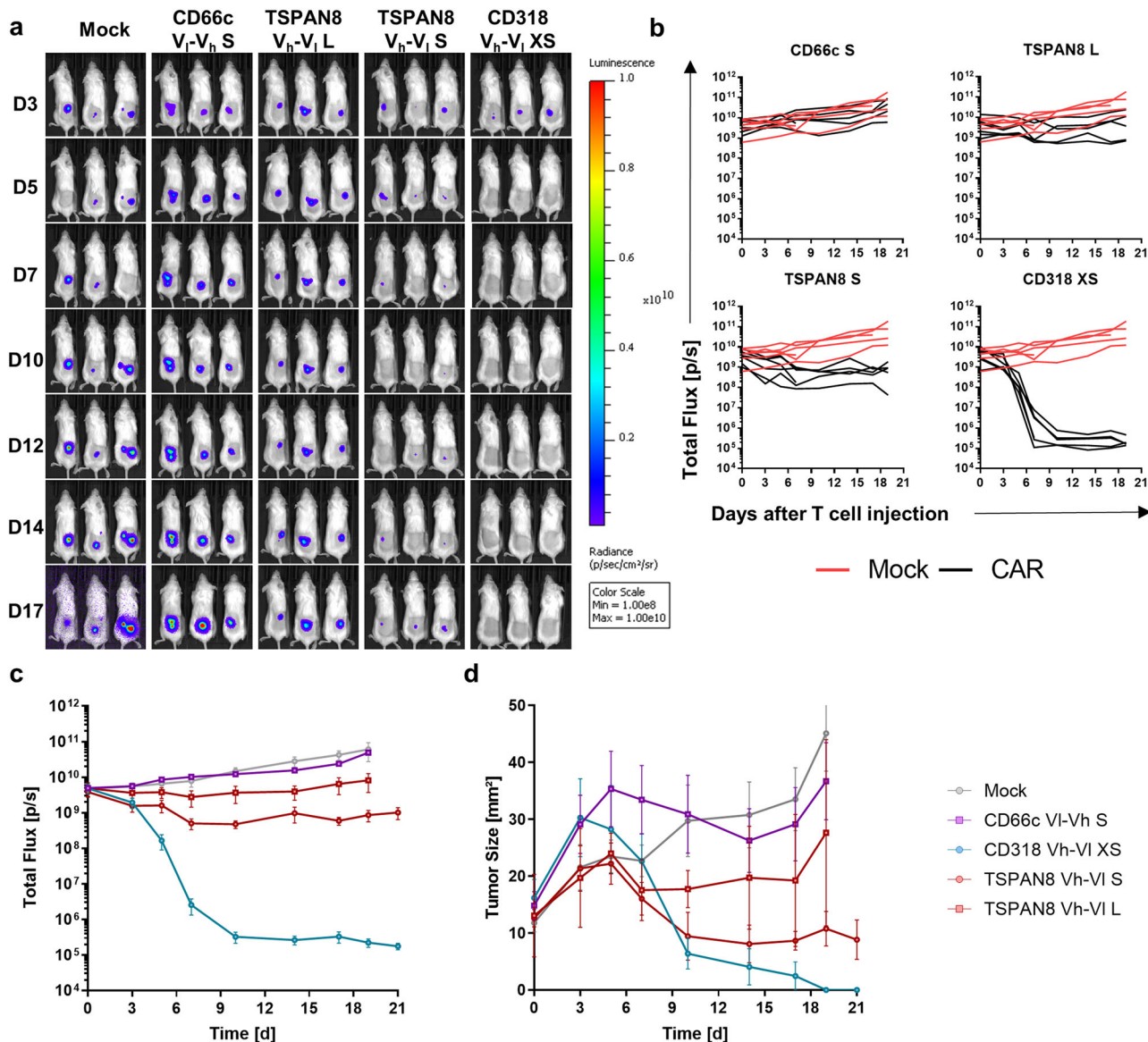

**Fig. 6 Evaluation of CAR T cell in vivo functionality in a BxPC3 xenograft model. a** Representative bioluminescence images of tumor-bearing NSG mice. Tumors were induced by subcutaneously transplanting luciferase expressing BxPC3 (color scale for all images, min = 1 × 10⁸, max = 1 × 10¹⁰). Mice were randomized and treated upon established solid tumors reached 25 mm² (day 15) by intravenous infusion of 1 × 10⁷ CAR T cells or Mock T cells. **b** Development of tumor burden for individual mice treated either with Mock T cells or with the respective CAR T cells (Mock: $n = 6$, CD66c S: $n = 6$, CD318 XS: $n = 6$, TSPAN8 S: $n = 6$, TSPAN8 L: $n = 6$). **c** Average bioluminescence ± s.e.m. of the respective treatment groups ($n$ equal to **b**). **d** Average tumor size ± s.e.m. of the respective treatment groups ($n$ equal to **b**). Source data are provided as a Source Data file.

was observed for the CD66c S CARs, correlating with the poor outcome in the first study. Reduced efficacy of the TSPAN8 CARs could be caused by reduced target expression of this tumor model as compared to the AsPC1 xenograft (Supplementary Fig. 5a). As observed for AsPC1, reduction of the tumor size was delayed probably as a result of T cell and macrophage infiltration (Fig. 7c). Analysis of the total T cell counts in the spleen at the end of the study confirmed that higher CAR T cell counts correlated with higher efficacy, again CD66c CAR T cells being the exception (Fig. 7a). T cell phenotyping revealed a decrease in the $T_{CM}$ compartment (Fig. 7b) within the CD66c S CAR T cells as compared to all other CAR-treated groups.

**Activation of CAR T cells specific for the target candidates is depended on target expression.** For CD318 and TSPAN8, the in vivo CAR T cell efficacy correlated with the target expression.

Albeit the CD318-directed CAR exhibited highest in vivo performance, CD318 was the target showing a heterogeneous expression on PDXs and primary tissues (Fig. 1a, c), raising concerns for its applicability among heterogeneous patient cohorts. Thus, we analyzed the ability of CAR T cell activation and cytokine release after 48 h on various dissociated PDX models, primary cells, and established cell lines showing heterogeneous target expression (Fig. 8a, b and Supplementary Fig. 11). We included the CD318 XS and TSPAN8 S spacer CAR, respectively, and both of the CD66c-directed XS and S spacer CARs, as the S spacer CAR did not evoke tumor shrinkage in the in vivo experiments. In addition, the CD66c XS spacer CAR showed superior release of IL-2 and other cytokines (Supplementary Data 7), which plays a crucial role in achieving the positive benefits of costimulatory signaling on T-cell survival, proliferation, and in vivo persistence[31]. PanCa0201 is a low

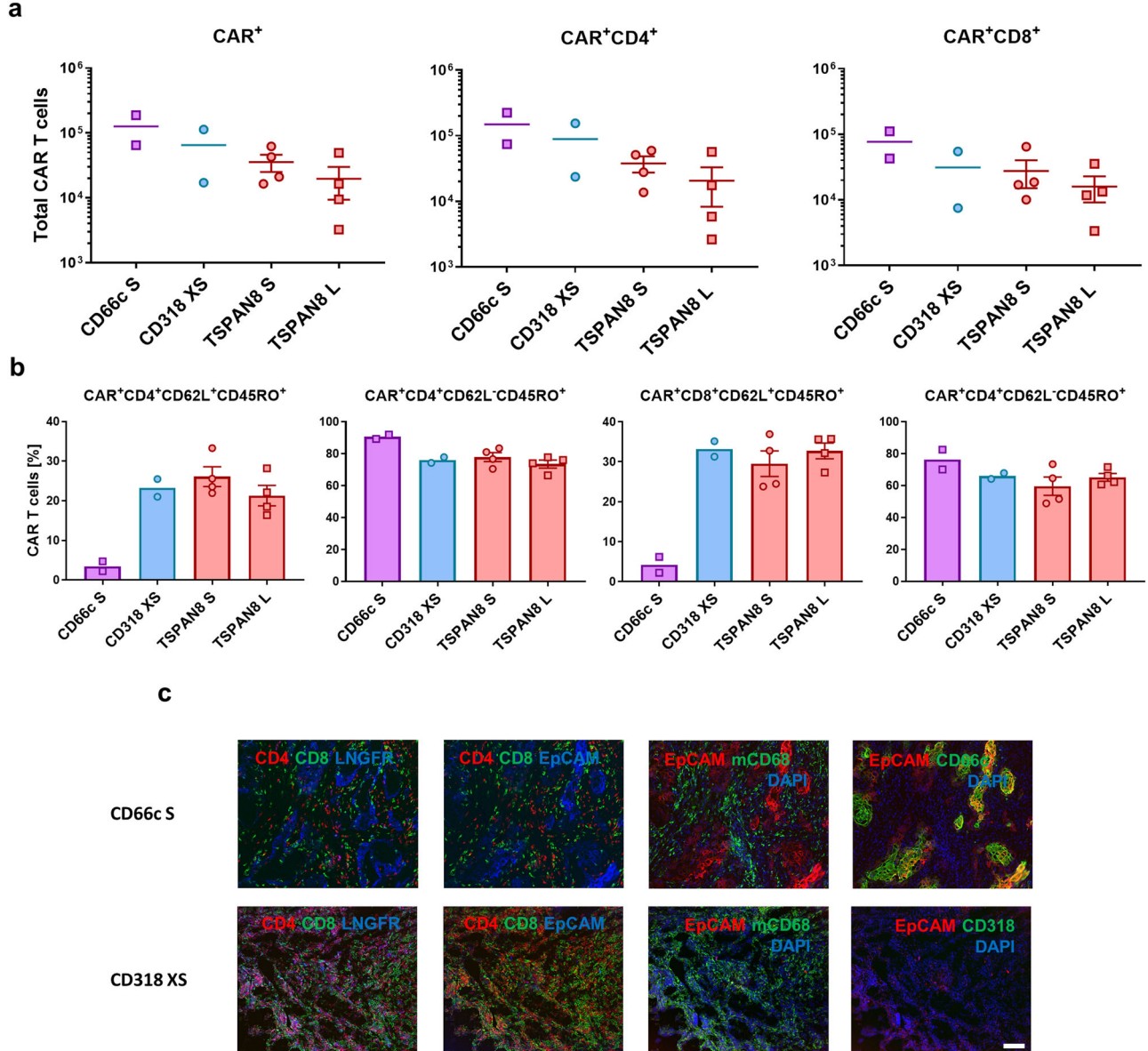

**Fig. 7 Ex vivo analysis of the T cell phenotype and AsPC1-derived tumor tissues upon treatment with CAR T cells. a** Number of CAR T cells in the spleen at the end of the experiment 25 days (CD66c S) and 27 days post CAR T cell injection (CD66c S: $n = 2$, CD318 XS: $n = 2$, TSPAN8 S: $n = 4$, TSPAN8 L: $n = 4$). **b** Phenotype of CAR T cells in the spleen at the end of the experiment 25 days (CD66c S) and 27 days post CAR T cell injection, as demonstrated by the percentage of $T_{CM}$ and $T_{EM}$ ($n$ equal to **a**). All data are shown as mean ± s.e.m. **c** Representative immunofluorescence images of (CAR) T cell tumor infiltration, macrophage tumor infiltration and target expression (CD318 XS tumor 10 days post CAR T cell injection, CD66c S tumor 25 days post CAR T cell injection). Staining was performed on one tumor of the respective treatment group and each image is representative for at least two regions of interest. Regions of interest during cyclic IF were chosen based on manual prestaining of DAPI and EpCAM. Scale bar = 100 μm. Source data are provided as a Source Data file.

passage cell line that we directly derived from a primary PDAC. It expressed all target candidates and could elicit high CAR T cell activation and cytokine release on all constructs (Fig. 8a, b). Using cells directly isolated from fresh PDX tumors showing heterogeneous target expression, only the CARs evolved T cell reactions when the respective target antigen was expressed. On PAXF1881, even the low amounts of TSPAN8 expression on only ~10% of the cells led to markedly elevated activation, which was not the case for CD66c CARs (Fig. 8a, b and Supplementary Fig 11). While this increases the chance of targeting tumors with low target expression, also healthy tissues with low target expression may be affected. Lower CAR T cell activation on PAXF cells as compared to cell lines could be caused by lower

target expression (Supplementary Fig. 11), and also by higher amount of debris and dead cells within the well when using freshly dissociated tissues which may hamper CAR T functionality. In these assays, the CD66c XS spacer CAR outperformed the S spacer CAR. Thus, we finally challenged both CD66c CARs in an AsPC1-based in vivo study (Fig. 8c–e). Indeed, the XS spacer CAR had a stronger and faster reaction leading to a strong and lasting tumor decrease in four out of five mice, proving the superiority of the XS spacer over the S spacer design when directed toward CD66c.

In conclusion, we could verify therapeutic efficacy of our CAR constructs at different therapeutic magnitudes ranging from tumor control to total tumor eradication and revealed an overall

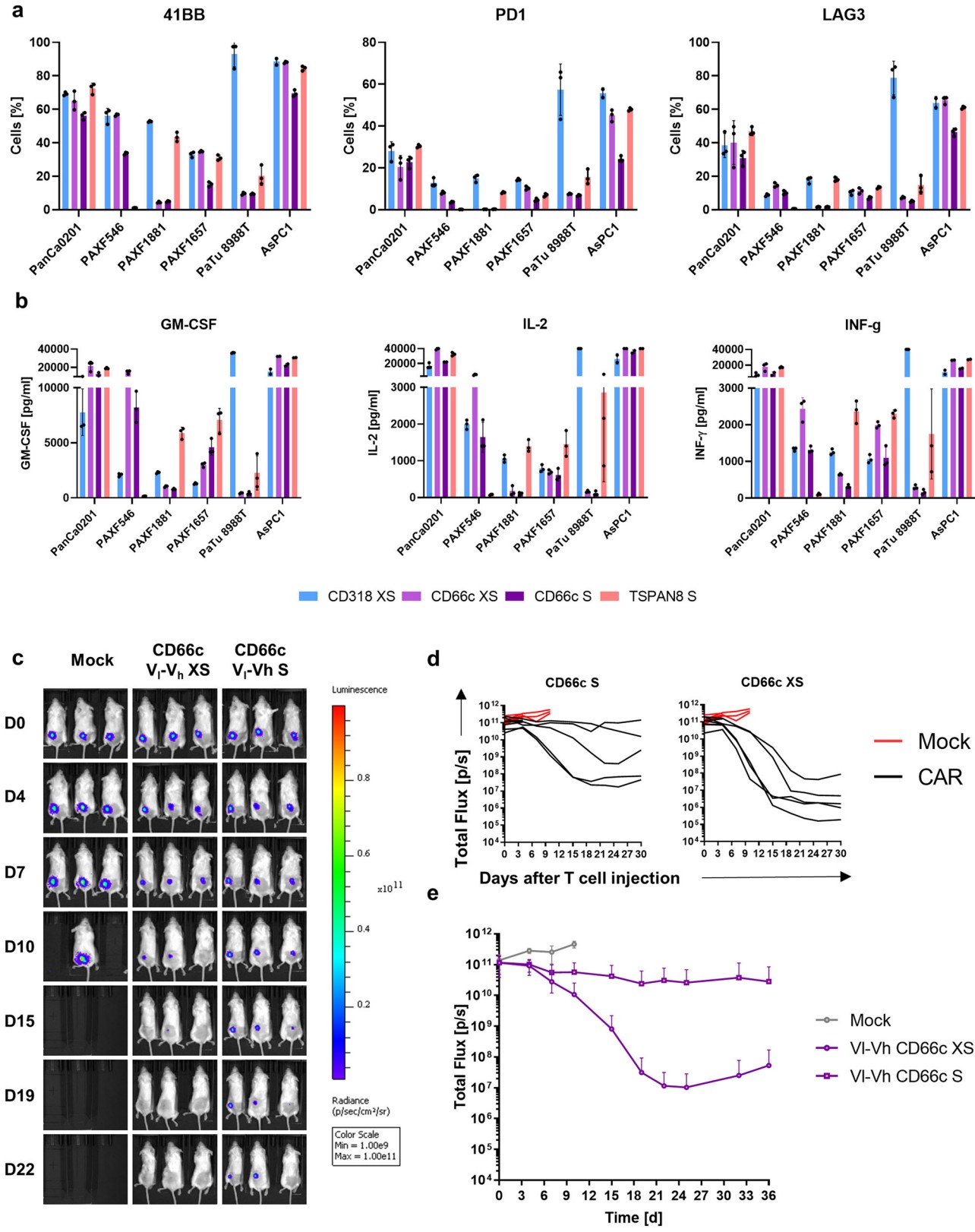

high correlation among in vivo studies. In addition, low-responsiveness—as seen for the CD66c construct—was not linked to target downregulation but rather to decreased CAR T cell infiltration or residency at the tumor.

**CD318 is the most promising candidate with respect to functionality and safety**. The bioinformatical analysis of CD66c,

CD318, and TSPAN8 showed that all of them have restricted but detectable expression on healthy tissues (Supplementary Data 2–4 and Supplementary Fig. 3), implying the need for an evaluation of safety concerns prior to a possible clinical translation. To overcome the inconsistency of publicly available databases we analyzed protein expression and localization by cyclic IF on 17 healthy tissues.

**Fig. 8 Target-specific activation of CAR T cells in vitro and evaluation of CD66c XS CAR in vivo. a** Activation marker expression on CAR T cells upon co-culture with target cells. **b** Cytokine release of CAR T cells upon co-culture with target cells. AsPC1 and PaTu 8988 T are established cell lines, PanCa0201 is a cell line below passage 10 derived from a primary PDAC and different PAXF refer to freshly dissociated human PDAC PDX tumors. **a**, **b** Shown are mean ± s.e.m ($n = 3$). E:T = 2:1. Activation marker expression and cytokine release patterns were measured after 48 h. **c** Representative bioluminescence images of tumor-bearing NSG mice. Tumors were induced by subcutaneously transplanting luciferase expressing AsPC1 (Color scale for all images, min = $1 \times 10^9$, max = $1 \times 10^{11}$). Mice were randomized and treated upon established solid tumors reached 25 mm² by intravenous infusion of $1 \times 10^7$ CAR T cells or Mock T cells. **d** Development of tumor burden for individual mice treated either with Mock T cells (red) or with the respective CAR T cells (Mock: $n = 5$, CD66c S: $n = 5$, CD66c XS: $n = 5$). **c** Average bioluminescence ± SD of the respective treatment groups ($n$ equal to **d**). Source data are provided as a Source Data file.

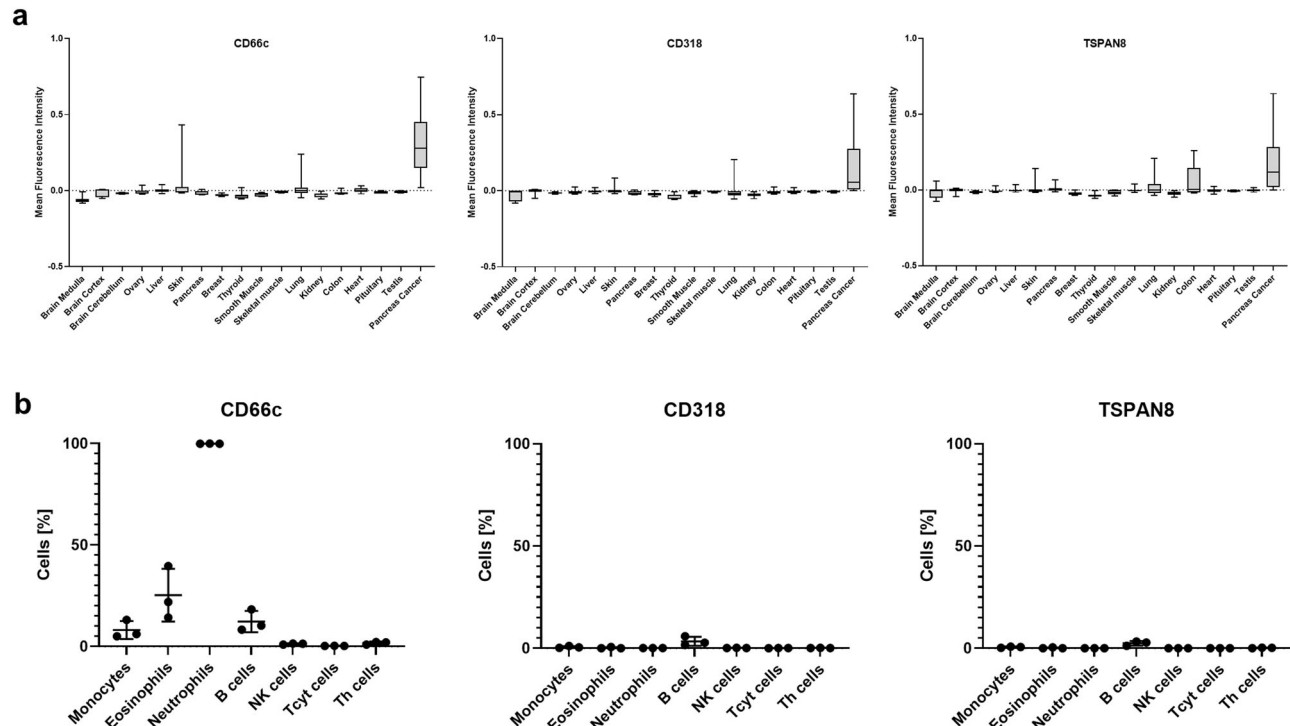

**Fig. 9 Off-tumor target expression. a** Quantification of target expression within different healthy tissues. Box-and-Whisker Plots show the distribution of the background corrected mean fluorescence intensity of identified cells in the respective tissues. The lower and upper hinges in the Box-and-Whisker Plots correspond to the first and third quartile (25th and 75th percentiles). The bar in the box depicts the median. Whiskers extend to the 5th and 95th percentile. CD66c: $n$ (left to right) = 4603, 1695, 716, 6591, 5746, 13301, 8682, 1731, 13391, 4066, 5878, 2966, 2737, 10844, 11595, 7405, 3671, 3371. CD318: $n$ (left to right) = 9341, 6542, 5282, 11438, 10593, 18148, 18395, 6578, 18238, 8913, 10725, 7813, 7584, 13933, 16442, 12252, 7112, 8218. TSPAN8: $n$ (left to right) = 9341, 6554, 5563, 11438, 10593, 18148, 18395, 6578, 18238, 8913, 10725, 7813, 7584, 13933, 16442, 12252, 7112, 8218. **b** Flow cytometric analysis of target expression on lysed blood samples. Data represent mean ± SD of three donors. Source data are provided as a Source Data file.

Quantification of protein expression supported our initial findings that all three target candidates showed a much stronger expression on PDAC tissue as compared to all healthy tissues analyzed. CD318 showed the most favorable pattern with almost no detectable protein expression in healthy tissues. The only relevant but very low expression was restricted to the luminal side of the colon, whereas T cells would most likely sense the basal or lateral membranes (Supplementary Fig. 12). TSPAN8 showed a strong expression in the gastrointestinal tract, especially on colon (Fig. 9a and Supplementary Fig. 13). This harbors the risk of toxicity induced by CAR T cells, whereas the weak staining observed in the medulla and skeletal muscle (Supplementary Fig. 13) seemed to derive from unspecific binding as the subtraction of the background signal resulted in a negative value (Fig. 9a). In case of CD66c, some expression was observed in the hair follicle (Supplementary Fig. 14) which might display a tolerable risk similar to expression detected in the ovary but limited to non-cellular mucus regions. A weak expression was

present in lung tissue but was much lower as compared to PDAC tissue (Fig. 9a), which might allow for a therapeutic window.

In addition, we measured expression of the three candidates on blood cells from healthy donors by flow cytometry as hematopoietic tissue displays a particular risk factor due to the presence of numerous putative target expressing cells in the preferred niche of CAR T cells. Flow cytometry-based analyses showed expression of CD66c but not of CD318 or TSPAN8 on several hematopoietic lineages (Fig. 9b). Whereas only minor subpopulations were CD66c-positive, the myeloid lineage, neutrophils in particular, showed a strong expression.

As all targets exhibited at least some minor somatic tissue expression, we finally investigated potential combinatorial targeting options that lead to T cell activation only when at least two targets are expressed[32–34]. Our cyclic IF multitissue array and an additional RNA expression analysis revealed the overall low-expression level of CD318, making it predestined as the main target for every combination (Fig. 9a and Supplementary Fig. 15).

It could then be combined with TSPAN8 or CD66c, without having substantial overlap of expression in any somatic tissue.

In conclusion, CD318 is the most promising candidate for a future clinical translation in terms of predicted safety and efficacy, while in the case of TSPAN8 and CD66c safety concerns arose with respect to gastrointestinal and hematopoietic tissues, respectively.

## Discussion

In the present study, we performed an empirical screening with the aim to discover CAR target candidates for the treatment of PDAC, identifying four target candidates: CLA, TSPAN8, CD66c, and CD318. We discovered CLA to be expressed on epithelial tumors. Until now, CLA was only known to be a unique skin-homing receptor, expressed on subsets of T cells, B cells, NK cells, Langerhans cells, monocytes, and dendritic cells[20,22–25]. It is a specialized glycosylated form of SELPLG and plays a role in tissue infiltration of immune cells through binding to E-, L-, and P-selectins[35]. Since it is only expressed on subsets of these cell types, we hypothesized that it might be a suitable CAR T cell target with acceptable off-tumor toxicities. However, as its expression was upregulated on activated T cells, it posed a technical roadblock producing functional CAR T cells. CLA-specific CAR T cells performed much worse underlining the intrinsic problem of CAR T cells specific for self-antigens. Other research groups developed potential solutions to this problem. While some suggested a CRISPR/Cas-mediated knock-out of the antigen within the T cells[36,37], others advocate to trap self-antigens in the T cells, using an ER retention domain coupled with an scFv specific for the self-antigen[38].

The other identified target candidates CD66c, CD318, and TSPAN8 have been described to be overexpressed in several cancer entities[39–46]. With the combination of our antibody-driven screenings and the comprehensive bioinformatical and IF analyses we could add the crucial knowledge that not only the expression of these target candidates is highly enriched in PDAC tumor cells but off-tumor expression within the human body is restricted as well. A strong support for the validity of this approach is that we identified targets which are already under investigation for CAR T cell-based treatment, such as CEACAM5 (CEA) or PROM1 (CD133)[9].

We evaluated 32 CAR constructs differing in scFv orientation and spacer length in vitro. While in the case of CD318 and CD66c the shorter spacer versions showed highest functionality, TSPAN8-specific constructs based on longer spacers performed better. CD318 has the longest extracellular amino acid chain (638), followed by CD66c (320) and TSPAN8 (24 and 96). Despite lacking knowledge on the exact epitope of the scFvs and the 3D structure of the extracellular domains, these findings nicely align with previous investigations pointing out that longer spacers are superior for membrane proximal epitopes and vice versa[47,48].

Finally, we could show that our four best-performing constructs in vitro also exhibited anti-tumor efficacy in two independent preclinical studies. The CD318 XS $V_h$-$V_l$ and the TSPAN8 S $V_h$-$V_l$ CARs turned out to be very promising candidates for further preclinical and possibly clinical evaluation based on their high efficacy. The CD66c S $V_l$-$V_h$ CAR was lagging behind its in vitro efficacy but could still induce disease stabilization in one of the two xenograft models. We could show that the decreased activity of the CD66c-specific CAR T cells was not caused by target downregulation and that the CAR T cell count in the spleen was indifferent to other treatment groups, therefore excluding persistence as an issue. At the tumor site, we found decreased macrophage and CAR T cell infiltration, suggesting

that infiltration and possibly activation of the CAR T cells may be impaired. Other studies have shown that tonic signaling can cause such differences among in vitro and in vivo studies[49]. However, we did not observe background activation of CAR T cells co-cultured with target-negative cells. When switching to the CD66c XS spacer-based construct, we found this one to be superior in vivo although only minor differences have been observed in vitro. One major difference was the superior release of IL-2 and other cytokines in case of the XS construct. It has been observed that the production of IL-2 indicates an appropriately functioning costimulatory domain. In addition, IL-2 plays a crucial role in achieving the positive benefits of costimulatory signaling on T-cell survival, proliferation, and in vivo persistence[31].

With respect to a possible clinical translation, we could corroborate the overlapping expression of CD66c and CEA, making it an interesting target candidate. In fact, it has been shown that CD66c is even more overexpressed in malignant tissues than CEA[50]. This finding might help to manufacture fine-tuned CARs that can differentiate among differential expression levels on tumor and healthy cells, as shown by Caruso et al. for the target EGFR[51]. However, the expression level of CD66c on neutrophils and other myeloid lineages underlines a strong risk for inducing a severe cytokine release syndrome (CRS) by massive CAR T cell reactivity in the blood as observed in B-ALL patients treated with CD19-specific CAR T cells, particularly observed in patients with high tumor burden[6]. However, while CRS and B cell aplasia can be treated quite well, neutropenia can only be tolerated in a very short time window rendering CD66c a high-risk target, at least in a single-specificity CAR T cell approach[52]. Combinatorial approaches using Boolean logic gating such as AND or NOT CAR constructs might offer an alternative to circumvent the expected toxicities[33,34,53]. We could show that a combination with CD318 might be the best option among our candidates. In this respect, also combinations of established targets such as MSN or CD133 and the targets described herein will broaden the repertoire to specifically target pancreatic tumor cells. Further alternatives would be the development of CARs tuned to distinguish among levels of antigen expression on tumor vs. healthy cells[51] or a tight control of time and dose of binder administration, such as anti-FITC-directed CAR-Ts or UNI-CARs[54,55]. In addition, spatial control of CAR expression or activation may add a further level of safety[56]. These approaches have the potential to overcome on-target off-tumor toxicities as seen with single target-specific CAR T cells.

CD318 and TSPAN8 have been suggested as targets for pharmacological or antibody-driven therapies[57–59]. While these studies show decreased tumor growth rates or disease stabilization in vitro and in vivo, our results showed complete tumor eradication, which is most likely owed to the superior cytotoxicity of T cells representing a "living drug". This has been shown in B-cell malignancies by the achievement of complete disease remission upon CD19-specific CAR T cell administration after failure of prior CD19-specific antibody therapy. However, as CAR T cells are more efficient than antibodies and their pharmacokinetics hard to predict and control, they have to be administered cautiously for potential off-tumor toxicities. Our analyses aiming at predicting tissues at risk suggested that CD318 has the most favorable pattern with almost no detectable protein expression in healthy tissues. Based on our results, the highest risk for TSPAN8-specific toxicities were predicted in the gastro intestinal tract, in particular for colon, probably similar to those observed for CEA-directed CARs[60]. Similar to CEA, using an alternative CAR T cell infusion route into the hepatic artery might be a solution to overcome this possible health threat[61]. In addition, the toxicity toward colon tissue could potentially be further reduced

using an AND CAR strategy, with CAR T cells only being activated when they face both CD318 and TSPAN8.

Of course, next to safety, also efficacy has to be shown in a clinical setting. We showed that only target-positive tumor cells induced T cell activation. The target candidates occasionally exhibited a heterogeneous expression pattern on PDX and primary PDAC, potentially reflecting the heterogeneous genetic background of PDAC[62–64]. However, this is indifferent to previous PDAC CAR targets[9]. This raises the need of accompanying diagnostic tools predicting treatment efficacy. Most likely, target expression measured on patient biopsies will play a central role in the future of this highly personalized approach and the cyclic IF method described herein could help stratifying the patients.

It has been shown for CD19-directed CAR therapies that, despite initial detection downregulation or loss of target expression can lead to relapse[65,66]. The risk of tumors developing therapy resistance by antigen escape is dependent on the biological function of the target molecule defining how easy it will be for the tumor to become independent of it. We did not directly analyze the probability of target cell loss as a resistance mechanism in this study, although we did not observe this effect in our in vivo models. However, due to the limited timespan of mouse experiments as compared to human tumor treatment, this might be missed, as it was the case for CD19. A strong body of previous studies proved the pivotal roles of CD66c, CD318, and TSPAN8 for cancer aggressiveness and dissemination[67–72]. However, if they are vital for tumor-cell survival, in particular for PDAC, remains unclear. This further highlights the need for availability of multiple targets allowing for combination therapies, thereby reducing the risk of target-negative relapse by increasing the selective pressure on tumor cells.

In summary, this study combines an empiric antibody-based flow cytometry screen with a cyclic IF imaging platform and a comprehensive bioinformatical and experimental evaluation for off-tumor expression to identify PDAC-specific cell surface markers. We identified four target candidates, CLA, CD66c, CD318, and TSPAN8, for a possible cellular immunotherapy of PDAC by CAR T cells. CARs specific for CD66c, CD318, and TSPAN8 showed functionality in vitro and in vivo, with CD318 being the most favorable candidate for a clinical translation.

## Methods

**Antibody screening on patient-derived xenografts.** All PDAC PDXs were obtained from Charles River Discovery Research Services Germany GmbH. PDX models were dissociated using the Tumor Dissociation Kit, human in combination with the gentleMACS™ Octo Dissociator with Heaters (both Miltenyi Biotec). Subsequently, mouse cells were depleted using the Mouse Cell Depletion Kit (Miltenyi Biotec). Resulting cell suspensions were analyzed using the MACS® Marker Screen, human (Miltenyi Biotec) a monoclonal antibody panel containing 371 pre-titrated antibodies with nine isotype controls, or candidate antibodies selected from this panel for subsequent screening steps (Supplementary Fig. 1). To differentiate two PDX samples in one measurement, one sample was stained with the CellTrace™ Violet Cell Proliferation Kit (Invitrogen™). All samples were measured on a flow cytometer.

**Bioinformatical data mining and ranking.** After assigning gene symbols to the 50 top PDX-based target candidates, the corresponding gene and protein expression data were retrieved from the following data sources:

Human Protein Atlas[26] (v15, https://www.proteinatlas.org/), ProteomicsDB[27] (https://www.proteomicsdb.org/), Human Proteome Map[28] (http://www.humanproteomemap.org/), Genotype-Tissue Expression (GTEx, https://gtexportal.org/) and Genevestigator[73].

*Human Protein Atlas (HPA): antibody-based.* Protein expression scores in HPA are based on immunohistochemical data manually scored with regard to staining intensity and fraction of stained cells. We determined the fraction of the 83 cell and tissue types listed in HPA assigned to the protein expression levels "not detected", "low/medium", and "high". The top 50 target candidates were sorted based on the fraction of "not detected" cell types in descending order. Within a group of candidates exhibiting the same fraction of not detectable cell types, candidates were

further sorted based on the fraction of "medium/high" cell types in ascending order.

*Human Protein Atlas (HPA): RNAseq.* At the time of data download, gene expression scores in HPA were based on FPKM values. We determined the fraction of the 32 tissue types listed in HPA assigned to the gene expression levels "not detected" (<1 FPKM), "low/medium" (1–50 FPKM), and "high" (>50 FPKM). The top 50 target candidates were sorted based on the fraction of "not detected" cell types in descending order. When the value for "not expressed" was set to a cut-off at 0.5 TPM, the rankings did hardly change. Within a group of candidates exhibiting the same fraction of not detectable cell types, candidates were further sorted based on the fraction of "medium/high" cell types in ascending order.

*ProteomicsDB (PDB): mass spectrometry-based.* Protein expression data were retrieved as log10 normalized iBAQ intensity values. We determined the fraction of the 66 tissue types listed in ProteomicsDB with missing protein expression values for the top 50 target candidates and ranked them in descending order. Only peptides unique to the target protein were considered.

*Human Proteome Map (HPM): mass spectrometry-based.* Protein expression data were retrieved as intensity levels scaled from 1 (lowest) to 10 (highest). We determined the fraction of the 17 adult tissue types and 6 cell types listed in HPM assigned to the arbitrarily defined protein expression levels "not detected/low" (levels 0–3), "medium" (levels 4–7), and "high" (levels 8–10). The top 50 target candidates were sorted based on the fraction of "not detected" tissues in descending order. Within a group of candidates exhibiting the same fraction of not detectable cell types, candidates were further sorted based on the fraction of "medium/high" cell types in ascending order. Only peptides unique to the target protein were considered.

*GTEx: RNAseq.* The data used for the analyses described in this manuscript were obtained from GTEx_Analysis_2016-01-15_v7_RNASeQCv1.1.8. We determined the fraction of the 30 tissue types listed in GTEx assigned to the median gene expression levels "not detected" (<1 TPM), "low/medium" (1–50 TPM), and "high" (>50 TPM). The top 50 target candidates were sorted based on the fraction of "not detected" tissues in descending order. When the value for "not expressed" was set to a cut-off at 0.5 TPM, the rankings did hardly change. Within a group of candidates exhibiting the same fraction of not detectable cell types, candidates were further sorted based on the fraction of "medium/high" cell types in ascending order.

*Genevestigator.* Gene expression data (log2 space) were extracted from the Genevestigator collection of microarray data (species: Homo sapiens, platform: Affymetrix Human Genome U133 Plus 2.0, sample status: healthy; root: cell type. Hierarchical categories with identical expression values were condensed). Two-dimensional hierarchical clustering was performed using MeV 4.9.0 (http://mev.tm4.org, distance metric: Euclidean distance, linkage method: complete linkage clustering; leaf order optimization: genes and samples).

*Rank sum.* The final rank was calculated based on the quotient of the rank sum from all data sources divided by the number of data sources it was found in.

*TCGA Research Network.* RNA sequencing data from PDAC were retrieved from the TCGA Research Network (https://www.cancer.gov/tcga). We sorted for our highest ranked target candidates (including CD66c, CLA, or SELPLG, respectively, CD318 and TSPAN8) and aligned their expression over all investigated PDAC tumor tissues, including one metastasis.

*Ranking according to Perna et al. (2017).* Perna and colleagues[29] developed an algorithm to identify expression of target candidates throughout the human body using HPA, PDB, and HPM as input databases. We recreated their method using publicly available datasets and a few methodological differences. We also log$_{10}$ transformed the data from HPM and PDB, but did not perform a temporary correction of HPM data for multiple gene assignments. Subsequently, normal distributions were fitted to the data using the Levenberg-Marquardt method. For some datasets this required ignoring data close to zero which we assume is noise introduced by various effects (for instance, genes never expressed in certain tissue types) and which did not fit in with a normal distribution. This curve fitting produced the average and standard distribution values we used and peak maximum and standard deviation were calculated for each curve.

Data were binned as follows: expression values between the maximum peak and one standard deviation above were considered "medium"[2]; values above this threshold were considered to be "high"[3] expressed; expression values between the peak maximum and the standard deviation below were considered "low"[1]; and all values below this threshold were categorized as "not detected"(0).

As the tissue names differed between the different data bases, they were harmonized to a consensus tissue name as suggested by Perna et al. (Supplementary Table 2). The final table was then created in depicting the highest value for the respective tissue and target candidate from all data bases.

*Weighing of vital and non-vital organs.* As substantial target expression in non-vital organs could be acceptable for CAR T cell therapy, we validated if our target candidates show different rankings when sorted according to vital or non-vital organ expression. For this, we categorized organs among these two groups according to Perna et al.[29]. We performed our ranking as described above. For Supplementary Data 3, only vital organ expression was considered, while non-vital organ expression was not taken into account. For Supplementary Data 4, we processed only non-vital expression, neglecting vital organ expression. In summary, both analyses reproduced our original ranking. To quantify disagreement between several rankings, we calculated Kendall rank correlation coefficients using R (v3.4) and the function "cor.test", e.g., "cor.test (S2, S3_1fpkm, method = "kendall")".

**Cyclic immunofluorescence staining.** Primary human PDAC specimens and healthy (normal adjacent) tissues were collected at the University Medical Center Göttingen from patients undergoing surgical resection of tumor masses. PDX have been sourced from in vivo studies within this paper. PDAC specimens, healthy tissues or PDX tissues were embedded in Tissue Freezing Medium (Leica) and stored at −70 °C until further use. Afterwards, three 8 μm sections were cut on a CM3050 S cryostat (Leica), collected on SuperFrost® Plus slides (Menzel) and stored no longer than 2 weeks at −70 °C. On the day of use, sections were thawed in −20 °C acetone and then processed for hematoxylin and eosin (HE) or immunofluorescence (IF) staining.

For HE staining, the acetone fixed section was dipped thrice in water and then incubated in Meyer's hematoxylin solution (Carl Roth) for 7 min. The section was transferred for 5 min into water and subsequently shortly dipped into HCl/ethanol solution (1% HCl, 70% ethanol). Afterwards, it was rinsed with water and stained in eosin/ethanol solution (0.5 g Eosin Y in 100 ml 70% ethanol) for 5 min. The section was then dehydrated stepwise first in 70% ethanol, then 96% ethanol, and finally pure ethanol with 3 dips for each concentration. Following this, the tissue was cleared in two changes xylene (Carl Roth) with three dips each. The HE-stained section was covered in Roti®-Histo Kit (Carl Roth) and cover slipped. Subsequently, the section was examined using a light microscope to define a region of interest, which was confirmed by a pathologist to be a neoplastic region.

After definition of a region of interest, another 8 μm section of the same specimen was thawed in −20 °C acetone. The fixed tissue was stored shortly in autoMACS™ Running Buffer (Miltenyi Biotec) and transferred to the MACSima™ Imaging Platform (Miltenyi Biotec). The MACSima™ Imaging Platform is a cyclic IF imaging platform enabling fully automated IF imaging of individual biological samples. The system operates by iterative fluorescent staining, image acquisition, and signal erasure, using multiple fluorochrome-labeled antibodies per cycle. Images were generated according to the manufacturer's instructions and analyzed using ImageJ 1.49v. A list of the used antibodies can be found in Supplementary Data 5.

**Healthy tissue microarray analysis and quantification of expression.** HDR images were checked manually using ImageJ 1.49v for any structures that could disturb the subsequent automated analysis. Therefore, images containing clumped conjugates, swollen nuclei, fabric remnants, or other artifacts were excluded from further analysis. All other images were uploaded into the CellProfiler v2.2.0 software[74]. We defined cell nuclei by DAPI staining as primary objects and excluded nuclei touching the image border. Secondary objects were defined as primary objects expanded by 5–6 pixels depending on the nuclei density, allowing us to approximate the shape of a cell. We then extracted the mean fluorescence intensities of the respective target candidates for each cell (secondary object). In order to define a background, we used the unspecific REA control (S) antibody as an isotype control and subtracted the control intensity from the target candidate intensity. For image display in the figures, we uploaded the respective PDAC image of each target to ImageJ and auto adjusted it. We then uploaded all other tissue images of the same excitation time and adjusted the display to the same parameters as for the PDAC image.

**Immunofluorescence and immunohistochemistry stainings of paraffin-embedded tissues.** Paraffin-embedded tissues of o.t. transplanted human AsPC1 and BxPC3 cells and human PDAC tissues and metastases (obtained from Proteogenex) were cut in 2–5 μm slices and collected on object slides. Subsequently, sections were either deparaffinized and stained with HE as described before[75] or deparaffinized and rehydrated sections were stained using IF or IHC. In brief, paraffin-embedded sections were cleared in three changes of xylene for 2 min. Sections were then rehydrated in three changes of 100% ethanol, one change of 95%, one change of 70%, and finally tap water. Each change took 2 min. Samples were then stained with HE as described above.

*IHC stainings.* IHC staining of o.t. CDX and metastases thereof were conducted using anti-CLA (clone HECA-452, Miltenyi Biotec, 130-091-634, 1/100), anti-CD66c (polyclonal, Abcam, ab199277, 1/100), anti-CD318 (polyclonal, Invitrogen, PA5-17245, 1/50), anti-TSPAN8 (polyclonal, Abcam, ab230488, 1/100), anti-rabbit horseradish-peroxidase (HRP) (polyclonal, Medac, 414142F), and Avidin-HRP (Invitrogen 18-4100-94). Sections were boiled for 20 min in Target Retrieval

Solution (Agilent) either with pH 6.0 for CLA, CD66c, and TSPAN8 or pH 9.0 for CD318 with subsequent incubation for 5 min in ice cold water and rinsing twice with TBS buffer for 5 min each. Afterwards, slices were incubated for 10 min with 3% hydrogen peroxide at RT followed by two 5 min washing steps with TBS. Unspecific antibody binding was averted by a 20 min blocking step with Fish block (SurModics). Primary antibody staining was performed overnight at 4 °C. Samples were washed and secondary antibody (anti-rabbit-HRP, undiluted or Avidin-HRP, 1:500) was added for 30 min (anti-rabbit-HRP) or 1 h (Avidin-HRP) at RT. Subsequently, samples were washed and enzyme substrate (AEC, BD Pharmingen, Cat. 551015) was added to the sections for 20 min at RT. Images were acquired using an Axioskop II microscope (Zeiss) and an Axiocam 105 color camera (Zeiss).

*IF stainings.* Human PDAC tissues and metastases thereof were stained with anti-CD66c (polyclonal, Abcam, ab199277, 1/100), anti-TSPAN8 (polyclonal, Abcam, ab230488, 1/100), anti-CD318 (polyclonal, Abcam, ab223743, 1/50), anti-cytokeratin-FITC (clone REA831, Miltenyi Biotec, 130-112-931, 1/10), and anti-rabbit-PE (clone 2A9, Southern Biotech, 4090-09, 1/125). Sections were boiled for 20 min and briefly washed in PEB buffer. Sections were then blocked using FcR Blocking Reagent, human (Miltenyi Biotec) for 15 min at RT. Slides were washed thrice and subsequently primary antibody staining as performed overnight at 4 °C. Again, samples were washed in PEB and cell nuclei were stained for 15 min at RT using DAPI Staining Solution (Miltenyi Biotec). Slices were washed again and secondary staining was conducted for 1.5 h at 4 °C using anti-rabbit-PE. Simultaneously, samples were incubated with the anti-cytokeratin-FITC antibody. Subsequent to this, samples were washed and slides were covered using Dako Fluorescence Mounting Medium (Agilent). Images were taken using an EVOS M5000 Imaging System (Thermo Fisher).

**Flow cytometric analysis of primary PDAC specimen.** Tumor tissues from seven PDAC patients undergoing surgical resection of tumor mass were collected at the University Medical Center Göttingen. The tumors were dissociated using the Tumor Dissociation Kit, human in combination with the gentleMACS™ Octo Dissociator with Heaters (both Miltenyi Biotec) and prepared for flow cytometry as described above.

**Cell lines and culture conditions.** Human embryonic kidney 293T cells (HEK293T, ACC 635) were obtained from the DSMZ—German Collection of Microorganisms and Cell Cultures. BxPC3 (CRL-1687) and AsPC1 (CRL-1682) cells were obtained from ATCC and cultured as recommended. For co-culture with T cells, they were transduced to express firefly luciferase (Luc) and green fluorescent protein (GFP). The PanCa0201 cell line was derived from a human primary PDAC biopsy, dissociated as described above and tumor cells were isolated prior to seeding using the Tumor Cell Isolation Kit, human (Miltenyi Biotec). Subsequently, tumor cells were cultivated using the Pancreas TumorMACS Medium (Miltenyi Biotec). The PanCa0201 cell line was transduced as well to express Luc and GFP. To validate authenticity of the cell lines used, we used the Human STR Profiling Cell Authentication Service (ATCC).

**Isolation of T cells and generation of CAR T cells.** Peripheral blood mononuclear cells (PBMCs) were isolated by density gradient centrifugation from buffy coats of healthy anonymous donors (German Red Cross Dortmund). T cells were purified from PBMCs using the Pan T Cell Isolation Kit, human (Miltenyi Biotec) and activated in TexMACS™ Medium (Miltenyi Biotec) containing T Cell Trans-Act™, human (Miltenyi Biotec) and 100 IU/ml of recombinant Human IL-2 IS, research grade (IL-2) (Miltenyi Biotec) or 12.5 ng/ml of recombinant human interleukin IL-7 and 12.5 ng/ml of recombinant human IL-15 (both Miltenyi Biotec) for the CLA expression experiments. T cells were transduced 24 h after activation using vesicular stomatitis virus glycoprotein G (VSV-G) pseudotyped lentiviral supernatants derived from transfected HEK293T cells. Supernatants were concentrated and stored at −70 °C until transduction. Three days post activation, T Cell TransAct™, human, was washed out of the medium and T cells were cultured with 10 IU/ml IL-2 containing TexMACS™ Medium. T cells were used for in vitro assays directly or frozen until further use for in vivo testing 12–14 days after purification from PBMCs. Frozen T cells that were used for in vivo testing were thawed 24 h before injection in TexMACS™ Medium without further supplements. On the day of use, the amount of living CAR T cells was determined using flow cytometry, and staining T cells with 7-AAD and anti-human low-affinity nerve growth factor receptor (LNGFR) (both Miltenyi Biotec).

**Generation of CAR plasmids.** Plasmids encoding the CAR constructs were prepared using standard molecular biology and cloning techniques. They all comprised murine single-chain variable fragments (scFvs) specific for the respective target candidates (CLA, CD66c, CD318, or TSPAN8) that were preceded by a CD8α leader peptide. Sequences for the light and heavy chain of the scFvs were the same as for the antibodies used during target discovery. scFvs were used in both possible orientations and connected with a glycine-serin linker. scFvs were followed by spacers differing in size. The library of backbones comprised either a human IgG4 hinge from the CH2-CH3 domains (L; long spacer; 228 aa), a human IgG4 hinge from the CH3 domain (medium spacer; 119 aa), a human IgG4 hinge from

the sequence between CH2 and CH1 domain (XS; extra short spacer; 12 aa), or a human CD8α spacer (S; short spacer; 45 aa). All IgG4 spacer domains contained a 4/2 NQ mutation in the CH2 domain as well as an S → P substitution in the hinge region in order to reduce FcR binding. All CARs shared the same CD8α trans-membrane domain and featured 4-1BB/CD3ζ-derived intracellular signaling domains. The CAR sequence was linked to a P2A sequence to induce co-expression of truncated LNGFR.

**Target-T cell co-culture assays**. Target cells were inoculated in duplicates in 96-well culture plates at densities of $1–5 \times 10^4$ cells per well in 100 μl of their respective culture medium. Directly thereafter or 24 h later, CAR$^+$ T cells were added in E:T ratios of either 5:1, 2:1, or 1:1. Final vessel volume equalled 200 μl. The amount of T cells in the Mock control was adjusted to the number of total T cells in the CAR group with the highest total cell count. Cytotoxicity was measured as a decrease in green surface area with an IncuCyte® S3 Live-Cell Analysis System (Sartorius) and the supplied software IncuCyte S3 (v2017A, v2018C, and v2019A). The surface area at the start of the experiment was considered 100% and the following decrease or increase in surface area was set into relation. At the end of co-culture, either after 48 h or 6 days, CAR T cells were analyzed with a flow cytometer for their expression of TIM-3 or 4-1BB, LAG-3, and PD-1. In addition, after 48 h 100 μl of medium was used for analysis of the cytokine release. In case co-culture continued, 100 μl of fresh target cell medium was added again. Cytokine content was measured with the MACSPlex Cytokine 12 Kit, human (Miltenyi Biotec).

**Normalization for in vitro killing, cytokine release, and marker expression for cross-comparability**. To better compare the effectivity of the CAR constructs between cell lines, assays, and donors, we normalized the readouts for each assay. We normalized the cytokine release of GM-CSF, IL-2, TNF-α, and INF-γ to the highest value for each cytokine and calculated from them an average, resulting in the cytokine value. We normalized to the highest number of CAR$^+$ cells expressing the respective marker, with the markers being PD-1, LAG-3, TIM-3, 4-1BB, and 4-1BB/LAG-3 double positive. Then, we calculated the average of the normalized values resulting in the marker value. For killing, we used the highest decrease after 48 h and 6 days in green surface area as value to normalize to. Again the average of 48 h and 6 days values resulted in the killing value. Balloon diagrams were created using R (v3.4) with the packages reshape2, ggplot2 and viridis.

**Preclinical mouse models**. All experiments were approved by the Governmental Review Committee on Animal Care in NRW, Germany or Lower Saxony, Germany and performed according to guidelines and regulations (Landesamt für Natur, Umwelt und Verbraucherschutz NRW, Approval number 84-02.04.2017.A320 and Niedersächsisches Landesamt für Verbraucherschutz und Lebensmittelsicherheit, Approval numbers 33.9-42502-04-13/1085 and 33.9-42502-04-14/1511). Mice were kept in IVC stations at room temperature with dark/light cycle of 12/12 h and humidity between 45% and 65%. For IHC stainings, orthotopic CDX from AsPC1 and BxPC3 cells were established as described previously[76]. In brief, anesthesia was given by intraperitoneal injection of ketamine (75–100 mg/kg) and xylazine (15–20 mg/kg). A laparotomy of around 1 cm in length was conducted and the pancreas was subsequently exposed by gentle pulling of the stomach. Then, $1 \times 10^6$ PDAC cells were injected in 15 μl PBS. The pancreas was then replaced into the abdominal cavity. The peritoneum was closed using continuous vicryl suture (Metric 1.5, Ethicon, Norderstedt, Germany) and the skin using interrupted suture. For CAR experiments, CDX were established by injecting $1 \times 10^6$ BxPC3 or AsPC1 cells s.c. in the right flank of NOD SCID gamma (NSG; NOD.Cg-PrkdcscidIl2rgtm1Wjl/SzJ) mice (Jackson Laboratory, Bar Harbor, USA, provided by Charles River). Once tumors reached 25 mm² as measured with a calliper, $5 \times 10^6$ or $1 \times 10^7$ CAR T cells were injected into the tail vein. The amount of injected Mock T cells was adjusted to the number of total T cells in the CAR group with the highest total cell count. Anti-tumor response was measured longitudinally using an in vivo imaging system (PerkinElmer, Waltham, USA) after intraperitoneal injection of 100 μl (30 mg/ml) D-Luciferin (Potasium Salt, LUCK-2G, GoldBio) using the supplied software Living Image (v4.7.3). All measures to secure the well-being of mice were executed following the relevant animal use guidelines and ethical regulations. Where possible, tumors were excised and cut in two halves. One half was embedded in Tissue Freezing Solution (Leica) and stored at −70 °C until further use. The other half was dissociated as stated above and T cells were analyzed using flow cytometry. Ex vivo analysis of the spleen was performed after dissociation using the gentleMACS™ Octo Dissociator with Heaters according to the manufacturers protocol (Miltenyi Biotec) and red blood cell lysis using Red Blood Cell Lysis Solution (Miltenyi Biotec) using a flow cytometer.

**Flow cytometry**. All samples were measured on a MACSQuant® Analyzer 10 and analyzed using the MACSQuantify™ Software v2.13.0, FlowJow v10.7.1 and Microsoft Excel for Windows 2016. Antibody conjugates that were used for surface marker expression on PDAC primary tissues, PDXs, and T cell analysis are listed in Supplementary Data 5. Gatings were performed as shown in Supplementary Fig. 2 and for the 8-Color Immunophenotyping Kit, human (Miltenyi Biotec) as suggested by the supplier, with the exception that instead of CD3-PE the respective target antigen was used. Stainings conducted with antibody conjugates from

Miltenyi Biotec were performed as recommended by the supplier. Therefore, conjugates were added to cell suspensions in a final ratio of 1:11 or 1:50 and incubated 10 min at 4 °C. Antibodies from Sigma-Aldrich and Thermo Fisher were applied at concentrations of 10 μg/ml and incubated for 30 min at 4 °C. Subsequent to the incubation, a washing step was performed adding PEB in an excess of 9 times the staining solution. Cells were spun down and resuspended in appropriate volumes of PEB and measured on a flow cytometer. Dead cells were excluded using PI or 7-AAD. A list of the used antibodies can be found in Supplementary Data 5.

**Statistics**. Unless otherwise specified, all graphs show the mean with error bars representing the standard error of the mean. Statistical comparisons between more than two groups were conducted by two-way ANOVA with $P$-value < 0.05 using GraphPad Prism 7. For all mouse experiments, the number of independent mice used is listed in the figure legend. For cytokine, marker, and in vitro cell killing experiments, at least $n = 2$ wells, and experiments were repeated at least twice.

**Ethical concerns**. For all studies using human primary tissue, written informed consent was obtained following the guidelines of the approved Universitätsmedizin Göttingen Review Board protocol.

Peripheral blood mononuclear cells (PBMCs) were isolated from buffy coats of healthy anonymous volunteers that were purchased from the German Red Cross Dortmund. All blood samples were handled following the required ethical and safety procedures.

All animal experiments were approved by the Governmental Review Committee on Animal Care in NRW or Lower Saxony, Germany and performed according to guidelines and regulations.

Healthy whole-blood samples were taken from voluntary healthy donors that gave their written consent before.

**Reporting summary**. Further information on research design is available in the Nature Research Reporting Summary linked to this article.

## Data availability

Publicly available data were retrieved from: Human Protein Atlas[26] (v15, https://www.proteinatlas.org/), ProteomicsDB[27] (https://www.proteomicsdb.org/), Human Proteome Map[28] (http://www.humanproteomemap.org/), Genotype-Tissue Expression (GTEx, https://gtexportal.org/), Genevestigator[73] and the TCGA Research Network (https://www.cancer.gov/tcga). Source data are available as a Source Data file. The remaining data are available within the Article, Supplementary information, or available from the authors upon request. Source data are provided with this paper.

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

## Acknowledgements

We want to thank Michail Knaul and Jutta Kollet for their help and advice regarding bioinformatical matters and Sandy Reiß, Lara Minnerup, Vera Dittmer, Abigail J. Deloria, Sabine Wolfgramm, Niels Werchau, and Simon Lennartz for excellent technical assistance.

## Author contributions

D.S. and O.H. wrote the manuscript, D.S., A.B., and O.H. designed the study, D.S. and L.N.K. conducted experiments for target discovery, D.S., G.T.H., and S.T. did the bioinformatical analysis, D.S. performed in vitro assays, D.S., W.A.R., J.H., J.B. and C.L. performed the in vivo studies, D.S. and D. Lock cloned the CAR library, M.L., D. Lenhard, J.S., D.J.A., and P.S. supplied materials and resources, J.M.G. and D.P.L. performed and analyzed IHC stainings on o.t. xenografts, J.M.G., D.P.L., F.A., A.K., A.B., C.H., D.E., and O.H. supervised the project, all authors discussed the data and reviewed the manuscript.

## Competing interests

D.S., L.N.K., S.T., G.T.H., D.A., W.A.R., J.H., J.B., C.L., D. Lock, A.K., C.H., D.E., A.B., and O.H. are employees of Miltenyi Biotec B.V. & Co. KG. M.L., D. Lenhard, and J.S. are employees of Charles River Discovery Research Services GmbH. All other authors declare no competing interests. There are patent applications pending related to this work.
