## [Peer Review File · Nature Communications]

REVIEWER COMMENTS

Reviewer #1 (Remarks to the Author): with expertise in pancreatic cancer and CAR-T

The development of safe and effective solid-tumor CAR T-cell therapy is constrained by several obstacles. Despite the promising response to checkpoint blockade agents and immunotherapy in some solid tumors, pancreatic cancer remains particularly difficult to achieve efficacy in, which has been evidenced in multiple phase 1 clinical trials. A major obstacle is the selection of suitable antigen targets, since targetable cell-surface antigens are most commonly self-antigens, raising the possibility of "on-target, off-tumor" toxicity to normal tissues. Effective and safe CAR T-cell targeting would eliminate tumors while sparing normal tissues.

In this study, the authors address the selection of novel target antigens for pancreatic cancer CAR T-cell therapy. The authors screened >300 commercially available antibodies by antibody staining of pancreatic adenocarcinoma patient-derived xenograft (PDX) models and a bioinformatics approach that maximizes predicted therapeutic potential and prioritizes safety. The authors then used primary pancreatic ductal adenocarcinoma tissues and normal-tissue microarrays to further validate safety and potential efficacy *in vitro* and *in vivo*.

The authors are to be commended for a rigorous exploration and validation of targetability and safety. This study establishes a robust example of a "first step" that other groups may follow when selecting a candidate target antigen for treatment targeting of antibody- or single-chain variable fragment (scFv)-based immune therapies.

The following comments may be used as critiques suggesting where the authors might incorporate a more informed understanding of CAR T-cell immunology to modify their conclusions and perhaps incorporate additional experiments to highlight important nuances to target antigen selection.

1. The authors have access to numerous PDX models, yet they use immortalized pancreatic cancer cell lines to validate *in vivo* functionality. This might lead to incorrect conclusions, such as, "CD318 is the most promising candidate with respect to functionality and safety." CD318 expression on primary pancreatic ductal adenocarcinoma tumor specimens is less than 50% on average, according to the authors' data, and there is a similar lack of uniform expression of CD318 on the PDX models tested. This raises the possibility that, although CD318 is a safe target, tumor antigen heterogeneity may limit the efficacy of a CD318-targeting CAR, resulting in a clinically ineffective treatment despite the authors' robust and meticulous selection strategy for tumor targeting. A more detailed understanding of antigen expression on primary tissues—both expression heterogeneity and levels of expression—would inform the therapeutic potential of selected CARs. *In vitro* and/or *in vivo* studies using primary tumor or PDX tissues would definitively address the authors' rather strong conclusion of CD318 candidacy and would perhaps indicate another target as the most efficacious, albeit with a less strong safety profile.

2. The authors test various spacer lengths and scFv orientations in *in vitro* functional assays. What is the rationale behind the equal weighting of cytotoxicity, activation marker expression, and cytokine expression in the CAR selection process, however? Many CARs do not produce IL-2, the production of which indicates an appropriately functioning costimulatory domain (in this case, 4-1BB); furthermore, IL-2 plays a crucial role in achieving the positive benefits of costimulatory signaling on T-cell survival, proliferation, and *in vivo* persistence. It is quite possible that a focus on IL-2 production would result in the selection of a different CAR with greater therapeutic potential, capable of tumor elimination at lower doses. The *in vitro* study that addresses costimulatory function and most accurately predicts CAR T-cell persistence and *in vivo* function has traditionally been repeated in *in vitro* antigen stimulation, which more closely approximates the stress on T cells posed by large tumor antigen burdens. Conducting similar *in vitro* antigen stimulation might inform the lack of CD66c functionality and may lead to the selection of different constructs for any of the target antigens.

3. A major shortcoming of the manuscript is the lack of discussion of the limitations of the approaches adopted for identifying the target antigen. In the identification of a candidate, differential antigen expression on cancer and normal tissue is only one aspect of defining a target. As there is no evidence in the authors' study that the identified antigen targets are crucial for cancer aggressiveness, cancer cells may readily shed the antigen and develop antigen escape. While I would not impose on the authors the need to cover all aspects of target antigen validation, including a concise description of the limitations of the proposed approaches will enhance the quality of an important study such as this one.

4. The authors' effort in generating CARs to demonstrate in vitro and in vivo efficacy of the target antigens is an important element in confirming targetability. The unfocused exercise of various spacer lengths and scFvs, however, is a bit of a stretch in the context of the manuscript, and it does not validate efficacy beyond targetability and safety. This should be addressed in the Discussion section.

Reviewer #3 (Remarks to the Author): with expertise in pancreatic cancer

The manuscript by Schaefer et al. takes step to tackle the significant issue of identifying relevant antigens in pancreatic cancer that could be used to develop CAR constructs for redirecting T cells. The authors conducted an extensive degree of screening of PDX tumors using flow cytometry and bioinformatics approaches, then validated these using a novel histology based approach. They designed CAR and transduced T cells to target CD66c, CD318, TSPAN8, putative novel antigens expressed in pancreatic cancer. Functional data are provided along with in vivo data demonstrating the ability of these newly generated CAR T cells to engage their target and have antitumor activity in vivo using standard models. Overall, the manuscript seems to have rigorous data, with a high level of transparency in the presentation, replication and methodology. There is a high level of innovation given the identification of novel targets that could be taken into subsequent steps for further testing. Their analysis of normal tissue expression of these targets is a further strength that will be relevant if constructs are ultimately able to be translated. There are a few comments below that could be used to enhance the quality and impact of the manuscript:

1. Introduction: The sentence: "These target candidates all have in common their shared expression on malignant and healthy tissues and toxicities in humans were already reported for HER2 and CEA" requires a reference(s).
2. Results: it would be helpful as a supplemental figure/table to know what mutational backgrounds are actually present in the PDX models used. This is stated, but not shown.
3. Results: Did the authors consider how anatomic location impacts expression of their candidate targets? The data are derived from subcutaneous PDX tumors in immune deficient mice. It would be important to know how the expression once in the actual pancreas or a metastatic site (i.e. liver) alters expression.
4. Results: the authors should look at expression of murine versions of their antigens in the classic KPC based models or cell lines. This would help the field.
5. Results: The majority of data are likely derived from specimens/PDX derived from resected PDAC tumors. Are there any data on these antigens in actual metastatic pancreatic cancer tissue or specimens (either fresh or from informatics databases)? This is important since CAR T therapy will likely be used in patients with metastatic disease, not primary PDAC tumors.

Typos:

Abstract, delete "a" from "the most promising candidate for a clinical translation..."

Intro, fix to read : "CAR T cells showed unprecedented efficacy' not "efficacies"

Methods, pg 21: need a space in between "sorted based"

Reviewer #4 (Remarks to the Author): with expertise in health bioinformatics

The main objective of the proposed bioinformatic approach for gene ranking is to determine, for each individual dataset (Human Protein Atlas, ProteomicsDB, Human Proteome Map and GTEx), how often a gene is "non-detected" across the studied tissue/cell types. The level of gene (or protein) expression is determined with pre-defined cut-off values. Perhaps, for gene expression, a gene with an expression level below 1 fpkm is consider "not detected". However, in literature, different cutoff values are suggested. The Expression Atlas database utilizes 0.5 FPKM or 0.5 TPM (<https://www.ebi.ac.uk/gxa/FAQ.html>) as minimum expression level. Does the final ranking change if 0.5 is used as cut-off for GTEx/RNA-seq data?

The defined computational procedure aims to assess/quantify the potential off-tumor effects when using immunotherapy with CAR-modified T cells. Despite CAR T cells being very promising, often toxicities are found with most of the clinical responses and, in some cases, fatal complications have been observed. In my opinion, when evaluating the safety aspect of treatments that could lead to severe cases of toxicity, one should also consider the tissue specificity and tissue-specific biological networks (<https://www.nature.com/articles/srep35241>). It has been shown that a disease in order to manifest itself in a particular tissue, a whole functional subnetwork of genes needs to be expressed in that tissue. Perhaps, a network-based approach could reveal important 'indirect' off-targets or the centrality of some specific of-targets.

What are the tissue-specific networks for pancreatic adenocarcinoma? What about using tissue-specific gene networks (built by integrating gene expression profiles and PPI networks) to identify indirect off-targets and to evaluate the 'network centrality' of off-targets? Should the studied tissues/cell types have different weights of importance? Perhaps, the rank sum could be weighted based on the relevance of tissues/cell types.

Overall, I suggest to implement a more robust bioinformatic approach to assess/quantify the potential off-tumor effects (PMC5932485)

REVIEWER COMMENTS

Reviewer #1 (Remarks to the Author): with expertise in pancreatic cancer and CAR-T

The development of safe and effective solid-tumor CAR T-cell therapy is constrained by several obstacles. Despite the promising response to checkpoint blockade agents and immunotherapy in some solid tumors, pancreatic cancer remains particularly difficult to achieve efficacy in, which has been evidenced in multiple phase 1 clinical trials. A major obstacle is the selection of suitable antigen targets, since targetable cell-surface antigens are most commonly self-antigens, raising the possibility of “on-target, off-tumor” toxicity to normal tissues. Effective and safe CAR T-cell targeting would eliminate tumors while sparing normal tissues.

In this study, the authors address the selection of novel target antigens for pancreatic cancer CAR T-cell therapy. The authors screened >300 commercially available antibodies by antibody staining of pancreatic adenocarcinoma patient-derived xenograft (PDX) models and a bioinformatics approach that maximizes predicted therapeutic potential and prioritizes safety. The authors then used primary pancreatic ductal adenocarcinoma tissues and normal-tissue microarrays to further validate safety and potential efficacy in vitro and in vivo.

The authors are to be commended for a rigorous exploration and validation of targetability and safety. This study establishes a robust example of a “first step” that other groups may follow when selecting a candidate target antigen for treatment targeting of antibody- or single-chain variable fragment (scFv)–based immune therapies.

The following comments may be used as critiques suggesting where the authors might incorporate a more informed understanding of CAR T-cell immunology to modify their conclusions and perhaps incorporate additional experiments to highlight important nuances to target antigen selection.

1. The authors have access to numerous PDX models, yet they use immortalized pancreatic cancer cell lines to validate in vivo functionality. This might lead to incorrect conclusions, such as, “CD318 is the most promising candidate with respect to functionality and safety.” CD318 expression on primary pancreatic ductal adenocarcinoma tumor specimens is less than 50% on average, according to the authors’ data, and there is a similar lack of uniform expression of CD318 on the PDX models tested. This raises the possibility that, although CD318 is a safe target, tumor antigen heterogeneity may limit the efficacy of a CD318-targeting CAR, resulting in a clinically ineffective treatment despite the authors’ robust and meticulous selection strategy for tumor targeting. A more detailed understanding of antigen expression on primary tissues—both expression heterogeneity and levels of expression—would inform the therapeutic potential of selected CARs. In vitro and/or in vivo studies using primary tumor or PDX tissues would definitively address the authors’ rather strong conclusion of CD318 candidacy and would perhaps indicate another target as the most efficacious, albeit with a less strong safety profile.

We thank the reviewer for bringing up this important point and apologize for not being precise enough on our use of PDX models and why we did not use them for additional *in vitro* and *in vivo* assays. The PDX models used in the screening have been purchased from Charles River. The material was shipped to our lab and subsequently used for flow cytometry based analyses. We were not allowed to expand these samples *in vitro* or *in vivo*. Unfortunately, we don’t have primary PDX models established nor the ethical approval to do so in our own labs. Only after the target candidate identification, we performed CAR testing, but for the above mentioned reasons we did not have these tissues available anymore. Nevertheless, we completely agree with the reviewer that CD318 exhibits heterogeneous expression on PDX and primary tissues and this has to be addressed in killing assays. To address this question, we ordered additional PDX tissues under an amended contractual exception allowing for *in vitro* assays, dissociated them and performed co-culture with CAR T cells

transduced with the most efficacious constructs. We used freshly dissociated PDX models with heterogeneous target expression (PAXF 546, PAXF 1881 and PAXF 1657), a cell line that we generated directly from primary PDAC tissue and which was below passage 10 when used in the assays (PanCa0201), a PDAC metastasis derived cell line only expressing CD318 (PaTu8988T) and the previously used AsPC1 cell line as a control. We incorporated the new results in Fig. 8 and Fig. S11. As expected, CAR T cells only showed activation when the respective target candidate was expressed on the tumor cells. These results confirm that CD318 may be the best target but only in case of expression in a particular patient. Moreover, the cell line PaTu8988T which was derived from a PDAC metastasis only expressed CD318 but not the other target candidates. Accordingly, it only evoked a response in CD318 specific T cells. To further evaluate the expression of our target candidates in a metastatic situation, we have sourced matched human primary and metastatic PDAC biopsies and performed IF analyses (new Fig. 2). CD318 was expressed on these primary tumors and on all metastases. We have adapted our results and discussion part accordingly, pointing out that patient stratification has to happen before treatment, as not all patients will have expression of all targets. However, this is indifferent to previous target candidates such as mesothelin or CEA and actually indicates the need of multiple targets being available.

2. The authors test various spacer lengths and scFv orientations in in vitro functional assays. What is the rationale behind the equal weighting of cytotoxicity, activation marker expression, and cytokine expression in the CAR selection process, however? Many CARs do not produce IL-2, the production of which indicates an appropriately functioning costimulatory domain (in this case, 4-1BB); furthermore, IL-2 plays a crucial role in achieving the positive benefits of costimulatory signaling on T-cell survival, proliferation, and in vivo persistence. It is quite possible that a focus on IL-2 production would result in the selection of a different CAR with greater therapeutic potential, capable of tumor elimination at lower doses. The in vitro study that addresses costimulatory function and most accurately predicts CAR T-cell persistence and in vivo function has traditionally been repeated in in vitro antigen stimulation, which more closely approximates the stress on T cells posed by large tumor antigen burdens. Conducting similar in vitro antigen stimulation might inform the lack of CD66c functionality and may lead to the selection of different constructs for any of the target antigens.

We highly appreciate the reviewers suggestion and reanalysed our CAR assays with a greater weight on IL-2 release. Indeed, we found that in *in vitro* assays the initially chosen CD66c S construct released lower amounts of IL-2 as compared to the CD66c XS construct. For the other target candidates, the previously chosen constructs were already the ones with the highest IL-2 release. This observation drove us to perform new *in vitro* and *in vivo* studies including the CD66c XS spacer based construct (new Fig. 8). Indeed, it turned out that the XS spacer construct was superior to the S spacer CAR and we adapted text and figures of the manuscript accordingly. In addition, we apologize for not being clear regarding the weighting of cytotoxicity, activation marker expression, and cytokine expression in the CAR selection process. We weighted killing, marker expression and cytokine release equally as it is known that all these parameters influence CAR T cell activity. However, which cytokine or marker steers what response to which extent is not completely understood, yet. While the reviewer is absolutely right about the pivotal role of IL-2, we would like to point out that GM-CSF for example can also elicit an intrinsic immune response as GVAX trials have showed. Thus, a CAR with low IL-2 but high GM-CSF release could still be beneficial for a therapy.

3. A major shortcoming of the manuscript is the lack of discussion of the limitations of the approaches adopted for identifying the target antigen. In the identification of a candidate, differential antigen expression on cancer and normal tissue is only one aspect of defining a target. As there is no evidence in the authors' study that the identified antigen targets are crucial for cancer aggressiveness, cancer cells may readily shed the antigen and develop antigen escape. While I would not impose on the authors the need to cover all aspects of target antigen validation, including a concise description of

the limitations of the proposed approaches will enhance the quality of an important study such as this one.

We thank the reviewer for stressing out this point. Antigen escape induced relapse for sure will be one of the most critical hurdles for long-term clinical success. We have included a new paragraph in the discussion section to address this point as well as the limitations of our approach.

4. The authors' effort in generating CARs to demonstrate in vitro and in vivo efficacy of the target antigens is an important element in confirming targetability. The unfocused exercise of various spacer lengths and scFvs, however, is a bit of a stretch in the context of the manuscript, and it does not validate efficacy beyond targetability and safety. This should be addressed in the Discussion section.

We highly appreciate this suggestion and have significantly shortened the respective paragraph. However, as we not only wanted to describe new target candidates but also provide a set of functional CAR constructs, we needed to show a proof-of-concept for targetability *in vitro* and *in vivo*. Therefore, we would like to leave in the data and key findings of CAR design for reasons of completeness and reproducibility.

Reviewer #3 (Remarks to the Author): with expertise in pancreatic cancer

The manuscript by Schaefer et al. takes step to tackle the significant issue of identifying relevant antigens in pancreatic cancer that could be used to develop CAR constructs for redirecting T cells. The authors conducted an extensive degree of screening of PDX tumors using flow cytometry and bioinformatics approaches, then validated these using a novel histology based approach. They designed CAR and transduced T cells to target CD66c, CD318, TSPAN8, putative novel antigens expressed in pancreatic cancer. Functional data are provided along with in vivo data demonstrating the ability of these newly generated CAR T cells to engage their target and have antitumor activity in vivo using standard models. Overall, the manuscript seems to have rigorous data, with a high level of transparency in the presentation, replication and methodology. There is a high level of innovation given the identification of novel targets that could be taken into subsequent steps for further testing. Their analysis of normal tissue expression of these targets is a further strength that will be relevant if constructs are ultimately able to be translated. There are a few comments below that could be used to enhance the quality and impact of the manuscript:

1. Introduction: The sentence: "These target candidates all have in common their shared expression on malignant and healthy tissues and toxicities in humans were already reported for HER2 and CEA" requires a reference(s).

We completely agree with the reviewer that references were required here and added them accordingly.

2. Results: it would be helpful as a supplemental figure/table to know what mutational backgrounds are actually present in the PDX models used. This is stated, but not shown.

We thank the reviewer for pointing this out. We initially thought it would suffice to reference the commercial source where this information can be downloaded. However, we agree that it is easier and more appropriate to include an additional supplementary table (new Table S1).

3. Results: Did the authors consider how anatomic location impacts expression of their candidate targets? The data are derived from subcutaneous PDX tumors in immune deficient mice. It would be important to know how the expression once in the actual pancreas or a metastatic site (i.e. liver) alters expression.

Indeed, this is a very interesting question. To address this point, we performed IHC, IF and flow cytometry stainings on subcutaneously versus orthotopically transplanted tumors. We included the new data into Fig. 2b, Fig. S4i and Fig. S5. In summary, the expression did not differ among subcutaneously and orthotopically implanted tumors, underlining the value of our target candidates. As most of these mouse models progress too fast to establish metastases, we could only analyze BxPC3 and AsPC1 based metastatic CDX models, this data is included in Fig. S4i. In addition, we did include an expression analysis on biopsies of human PDAC metastases (liver and lymph node) indicating expression of our target candidates on metastatic lesions similar to the primary tumors (new Fig. 2a).

4. Results: the authors should look at expression of murine versions of their antigens in the classic KPC based models or cell lines. This would help the field.

We completely agree that it would be ideal to have genetically induced or syngeneic transplanted mouse models to get further insights into the biology of our target candidates. We followed the reviewers suggestion and expanded our IHC analysis also on KPC mice and a mouse PDAC cell line. Only for TSPAN8, a weak staining could be detected (data not shown). However, we would like to stress out, that from our 4 target candidates only TSPAN8 and CDCP1 have murine homologues. In

addition, the general expression patterns for these are very different among mouse and human tissues (biogps.org).

5. Results: The majority of data are likely derived from specimens/PDX derived from resected PDAC tumors. Are there any data on these antigens in actual metastatic pancreatic cancer tissue or specimens (either fresh or from informatics databases)? This is important since CAR T therapy will likely be used in patients with metastatic disease, not primary PDAC tumors.

We thank the reviewer for raising this important aspect and agree that the expression on metastases is of utmost importance, especially in the light of the mentioned prospect, that cellular therapies may most likely be applied in late – and thus metastatic – stages.

Metastases appear in disease stages where surgery is not applied anymore, making it challenging to access such tissues. Nonetheless, we were able to source three matched tissue pairs of primary and metastatic lesions and perform IF stainings (new Fig. 2a)). We could confirm expression of CD66c, CD318 and TSPAN8, indicating that CAR T cell therapy based on these may also be suitable to target metastatic lesions. In addition, we found our targets expressed in a dataset from the TCGA Research Network (<https://www.cancer.gov/tcga>) shown in new Table S6. Along this point and also in response to reviewer #1, we have adapted our discussion part being more careful and pointing out that patient stratification has to happen before treatment, as not all patients will show expression of all targets. However, this is indifferent to previous target candidates such as mesothelin or CEA and actually indicates the need of multiple targets being available.

Typos:

Abstract, delete “a” from “the most promising candidate for a clinical translation...”

Intro, fix to read : “CAR T cells showed unprecedented efficacy’ not “efficacies”

Methods, pg 21: need a space in between “sorted based”

We are thankful the reviewer pointed out these mistakes and corrected them accordingly.

Reviewer #4 (Remarks to the Author): with expertise in health bioinformatics

The main objective of the proposed bioinformatic approach for gene ranking is to determine, for each individual dataset (Human Protein Atlas, ProteomicsDB, Human Proteome Map and GTEx), how often a gene is “non-detected” across the studied tissue/cell types. The level of gene (or protein) expression is determined with pre-defined cut-off values. Perhaps, for gene expression, a gene with an expression level below 1 fpkm is consider “not detected”. However, in literature, different cutoff values are suggested. The Expression Atlas database utilizes 0.5 FPKM or 0.5 TPM (<https://www.ebi.ac.uk/gxa/FAQ.html>) as minimum expression level. Does the final ranking change if 0.5 is used as cut-off for GTEx/RNA-seq data?

We thank the reviewer for bringing up this important point. During the course of our study, we thoroughly discussed the cut-offs to be applied for the different databases as the decision among “low” and “no” expression could theoretically change the whole outcome of our ranking. We oriented ourselves on the available literature (in which there is no consensus as already mentioned by the reviewer), as well as our own experience regarding this matter. To address the impact of a different cut-off, we followed the reviewers suggestion and implemented new cut-off levels for different databases. We included the additional rankings into Table S3 and S4 and referenced it in the manuscript. When GTEx data was set to a cut-off at 0.5 FPKM (also applied for human protein atlas derived data), the rankings did hardly change, nicely confirming that the way of ranking has some inherent robustness.

The defined computational procedure aims to assess/quantify the potential off-tumor effects when using immunotherapy with CAR-modified T cells. Despite CAR T cells being very promising, often toxicities are found with most of the clinical responses and, in some cases, fatal complications have been observed. In my opinion, when evaluating the safety aspect of treatments that could lead to severe cases of toxicity, one should also consider the tissue specificity and tissue-specific biological networks (<https://www.nature.com/articles/srep35241>). It has been shown that a disease in order to manifest itself in a particular tissue, a whole functional subnetwork of genes needs to be expressed in that tissue. Perhaps, a network-based approach could reveal important ‘indirect’ off-targets or the centrality of some specific of-targets.

What are the tissue-specific networks for pancreatic adenocarcinoma? What about using tissue-specific gene networks (built by integrating gene expression profiles and PPI networks) to identify indirect off-targets and to evaluate the 'network centrality' of off-targets?

We appreciate the reviewers idea of implementing the tissue specific networks of PDAC in our research. It is a nice approach to enlighten why certain diseases appear only in specific tissues and detriment is not observed in all somatic tissues.

In our case however, this method cannot be applied to assess off-tumor toxicity. Our CAR constructs contain well established spacer domains, for which it has been shown that no (or neglectable) binding within the human body is happening. The scFv's may bear some unspecific binding besides their antigen specificity. Unfortunately, there is no way to estimate this bioinformatically and the reason we had to experimentally address this point by binding studies on healthy human tissues. Thus, we cannot evaluate off-target (meaning CAR binding to an unspecific molecule in our case) binding *in silico*. We can only evaluate on-target/off-tumor (meaning the CAR binds specific to its antigen, which is however expressed offside the tumor) binding *in silico*. This depends on the surface expression as CAR T cells can only become cytotoxically active, when the target antigen is presented to the extracellular compartment. Cell surface expression in turn cannot be predicted safely by analysing gene expression as post-translational processes as well as surface stability influence this to a great extent, making it difficult to predict, which gene product can be found at which level on the cell surface. As this by far is the most relevant measure for T cell engagement, surface expression had to be evaluated experimentally.

Should the studied tissues/cell types have different weights of importance? Perhaps, the rank sum could be weighted based on the relevance of tissues/cell types.

We completely agree with the reviewer and thought of this option. We initially decided against a weighted ranking for two reasons: 1.) There is no consensus in the scientific community about what makes a tissue dispensable. 2.) Even if a tissue would be dispensable, a massive T cell reaction could be provoked, developing into a severe and sometimes fatal cytokine release syndrome, as seen for B cell specific targets which are only expressed on dispensable cell lineages.

Nonetheless, we highly appreciate the idea and followed the reviewer's suggestion. We added new tables S3 and S4 in which we performed our ranking split between "essential" and "non-essential" tissues. We followed the decision of which organs are essential as proposed by Perna *et al.* (2017) as this seemed the most accepted approach. In summary, the overall ranking of our target candidates did not change strongly (only +/- 1 rank). Interestingly, established targets such as mesothelin moved down to a greater extent (-6 for MSLN) indicating a good level of confidence for the proposed new candidates.

Overall, I suggest to implement a more robust bioinformatic approach to assess/quantify the potential off-tumor effects (PMC5932485).

We thank the reviewer for suggesting a more sophisticated approach according to Sun *et al.* (2018). The referenced paper deals with the different kinds of toxicities that may arise during CAR T cell therapy. As discussed above, the only bioinformatically addressable toxicity was the on-target/off-tumor toxicity. In this reference, a so called "AND" CAR approach is suggested, meaning full CAR T cell activation is only happening in case two targets are present on the same target cell. To address this point, we analysed presence of our target candidates with a particular focus on their combined expression in a given tissue and identified combinatorial options with a high confidence of predicted safety. We adapted the manuscript accordingly and added novel Figure Fig. S15.

REVIEWERS' COMMENTS

Reviewer #1 (Remarks to the Author):

Authors appropriately addressed reviewer's concerns and answered appropriately. The revised manuscript shows improved interpretation of results and appropriately reflects on the excellent work performed.

Reviewer #3 (Remarks to the Author):

This revised manuscript has done a very thorough job at addressing the prior concerns from reviewers. In particular the authors have put forth a significant number of new experiments and made an even more comprehensive understanding of these potential new targets. Overall this will inform the field and improve options to pursue for cellular immunotherapy in PDAC. Only one small typo below should be corrected as below:

Page 8, first paragraph, last sentence, "extend" should be corrected to "extent".

Reviewer #4 (Remarks to the Author):

The authors have addressed my original comments in the significantly revised version.

Just a small note: it is evident that the rankings did hardly change. However, I suggest to quantify the disagreement between the two ranking lists by using the Kendall tau rank (or other similar metrics).

REVIEWER COMMENTS

Reviewer #1 (Remarks to the Author):

Authors appropriately addressed reviewer's concerns and answered appropriately. The revised manuscript shows improved interpretation of results and appropriately reflects on the excellent work performed.

We thank the reviewer for the helpful comments and positive feedback.

Reviewer #3 (Remarks to the Author):

This revised manuscript has done a very thorough job at addressing the prior concerns from reviewers. In particular the authors have put forth a significant number of new experiments and made an even more comprehensive understanding of these potential new targets. Overall this will inform the field and improve options to pursue for cellular immunotherapy in PDAC. Only one small typo below should be corrected as below:

Page 8, first paragraph, last sentence, "extend" should be corrected to "extent".

We thank the reviewer for the positive feedback. We corrected the typo accordingly.

Reviewer #4 (Remarks to the Author):

The authors have addressed my original comments in the significantly revised version.

Just a small note: it is evident that the rankings did hardly change. However, I suggest to quantify the disagreement between the two ranking lists by using the Kendall tau rank (or other similar metrics).

We appreciate the reviewer's feedback and thank the reviewer for the suggestion. Indeed, the Kendall rank correlation coefficient offers a more objective measure for differences in rankings. Thus, we gladly followed the reviewer's suggestion and calculated Kendall rank correlation coefficients for all rankings and implemented them as an additional sheet in the new Supplementary Data 3 and 4 (former Supplementary Tables S3 and S4).